# The Fml1-MHF complex suppresses inter-fork strand annealing in fission yeast

Io Nam Wong[1†‡], Jacqueline PS Neo[1†§], Judith Oehler[1†], Sophie Schafhauser[1], Fekret Osman[1], Stephen B Carr[2], Matthew C Whitby[1]*

[1]Department of Biochemistry, University of Oxford, Oxford, United Kingdom; [2]Research Complex at Harwell, Rutherford Appleton Laboratory, Harwell, United Kingdom

**Abstract** Previously we reported that a process called inter-fork strand annealing (IFSA) causes genomic deletions during the termination of DNA replication when an active replication fork converges on a collapsed fork (Morrow et al., 2017). We also identified the FANCM-related DNA helicase Fml1 as a potential suppressor of IFSA. Here, we confirm that Fml1 does indeed suppress IFSA, and show that this function depends on its catalytic activity and ability to interact with Mhf1-Mhf2 via its C-terminal domain. Finally, a plausible mechanism of IFSA suppression is demonstrated by the finding that Fml1 can catalyse regressed fork restoration in vitro.

**\*For correspondence:**
matthew.whitby@bioch.ox.ac.uk

[†]These authors contributed equally to this work

**Present address:** [‡]Faculty of Medicine, Macau University of Science and Technology, Taipa, Macau; [§]DSO National Laboratories, Singapore, Singapore

**Competing interests:** The authors declare that no competing interests exist.

## Introduction

Problems arising during DNA replication are a major cause of disease-promoting mutations and genome rearrangements (*Aguilera and García-Muse, 2013*; *Carvalho and Lupski, 2016*; *Cortez, 2019*; *Gaillard et al., 2015*; *Macheret and Halazonetis, 2015*). One especially challenging problem occurs when a replication fork encounters a barrier in the template DNA (e.g. a DNA lesion or tightly bound protein) that triggers its collapse (*Lambert and Carr, 2013*). Collapsed forks require the intervention of homologous recombination proteins to resume DNA synthesis through a process termed recombination-dependent replication (RDR) (also called break-induced replication [BIR] if the collapsed fork is broken) (*Ait Saada et al., 2018*; *Sakofsky and Malkova, 2017*). RDR is thought to prevent mitotic catastrophes by ensuring that genome duplication is completed before chromosomes are fully segregated during mitosis (*Özer and Hickson, 2018*). However, this beneficial function of RDR is somewhat offset by its proclivity to cause mutations and genome rearrangements (*Ait Saada et al., 2018*; *Sakofsky and Malkova, 2017*). Completion of DNA replication by an oncoming canonical fork, converging with the collapsed fork, limits the need for RDR and, therefore, reduces the frequency of mutations and genome rearrangements that would otherwise be caused (*Jalan et al., 2019*; *Mayle et al., 2015*; *Nguyen et al., 2015*). However, this type of fork convergence, involving an active and collapsed fork, is not without its own risks as it can lead to genomic deletions via a process called inter-fork strand annealing (IFSA) (*Morrow et al., 2017*).

Our favoured model for IFSA posits that the collapsed replication fork undergoes regression followed by Exo1-dependent resection to form a fork with a protruding 3' single-stranded (ss) DNA tail (*Morrow et al., 2017*). This is followed by the Rad52 recombination protein binding the tail and annealing it to complementary ssDNA exposed in the lagging strand gap of an oncoming replication fork. If this annealing occurs between repetitive DNA elements, then cleavage of the resulting 'IFSA junction' by a DNA structure-specific nuclease, such as Mus81, causes the formation of a genomic deletion (*Morrow et al., 2017*). Intriguingly, it was found that Fml1 plays an important role in suppressing genomic deletions induced by replication fork collapse at the *RTS1* replication fork barrier (RFB) in fission yeast (*Morrow et al., 2017*). However, it was not established whether these were

deletions coming from Rad52-mediated IFSA or from a pathway involving D-loop formation by Rad51 catalysed strand invasion (*Morrow et al., 2017*; *Nguyen et al., 2015*).

Fml1 is a member of the FANCM family of DNA helicases/translocases, which play a variety of roles in DNA metabolism (*Basbous and Constantinou, 2019*; *Whitby, 2010*; *Xue et al., 2015b*). The importance of FANCM in humans is highlighted by the fact that it both promotes fertility and acts as a potent tumour suppressor (*Basbous and Constantinou, 2019*). Many of the roles attributed to FANCM family members have been linked to their ability to catalyse branch migration of DNA junctions by harnessing the motor activity of their conserved N-terminal DEAH helicase domain. For example, they catalyse the dissociation of D-loops to drive DNA double-strand break (DSB) repair by synthesis-dependent strand annealing, and regress stalled replication forks to promote lesion bypass by template switching (*Crismani et al., 2012*; *Gari et al., 2008a*; *Gari et al., 2008b*; *Lorenz et al., 2012*; *Nandi and Whitby, 2012*; *Prakash et al., 2009*; *Romero et al., 2016*; *Sun et al., 2008*). FANCM family proteins also have a non-catalytic C-terminal domain that variously controls and directs their functioning through interaction with other proteins. One of these interactions, which is conserved from yeast to human, is with the centromeric proteins Mhf1 and Mhf2 (also known as CENP-S and CENP-X) (*Singh et al., 2010*; *Yan et al., 2010*). These small histone-fold proteins combine to form a structure that resembles the histone H3/H4 tetramer, which acts to support and modulate FANCM activity (*Bhattacharjee et al., 2013*; *Fox et al., 2014*; *Nishino et al., 2012*; *Singh et al., 2010*; *Tao et al., 2012*; *Wang et al., 2013*; *Xue et al., 2015a*; *Yan et al., 2010*; *Yang et al., 2012*; *Zhao et al., 2014*).

Although many roles have been attributed to FANCM and its orthologues, the finding that Fml1 suppresses deletions that might stem from IFSA represented a potentially novel function for this family of proteins (*Morrow et al., 2017*). In this paper we confirm that Fml1 does indeed suppress Rad52-mediated IFSA. Moreover, we establish that this role is dependent on its DNA motor activity and is promoted by Mhf1-Mhf2 in a manner that is dependent on its interaction with Fml1's C-terminal domain. Finally, we reveal a new in vitro activity for Fml1 (and, therefore, potentially for other members of the FANCM family too), namely catalysing the restoration of a regressed replication fork, which we speculate could account for how it suppresses IFSA in vivo.

## Results

### Fml1's catalytic activity and C-terminal domain are required for it to suppress *RTS1*-induced spacer-dependent deletions

To investigate Fml1's role in supressing IFSA, we used a recombination reporter consisting of a direct repeat of *ade6*⁻ heteroalleles, with an intervening *his3*⁺ gene and *RTS1* RFB, inserted at the *ade6* locus on chromosome 3 (*Figure 1A*). *RTS1* is a polar RFB meaning that it only blocks replication forks in one direction and at the reporter it is positioned in its so-called 'active orientation' (AO), blocking forks moving from telomere to centromere, which is the prevailing direction of replication at this genomic site (*Nguyen et al., 2015*). Replication fork blockage at *RTS1* strongly induces recombination between the *ade6*⁻ heteroalleles leading to two types of *ade6*⁺ recombinants, namely gene conversions and deletions (*Figure 1A*). The standard version of our recombination reporter has ~3 kb of DNA separating *ade6-L469* and *his3*⁺ genes (*Ahn et al., 2005*; *Nguyen et al., 2015*). However, we showed previously that expanding this distance, by insertion of spacer DNAs, results in a dramatic increase in the frequency of deletions (so-called spacer-dependent deletions or SDDs), which we proposed stemmed from IFSA (*Morrow et al., 2017*). Therefore, to investigate Fml1's role in suppressing IFSA, we used a version of the reporter with an extra 2 kb spacer and compared the frequency of Ade⁺ recombinants in a wild-type and *fml1Δ* strain (*Figure 1A,B*). Consistent with our previous finding (*Morrow et al., 2017*), we observed a ~ 3 fold reduction in gene conversions and ~2.5 fold increase in SDDs in a *fml1Δ* mutant. To see if Fml1's ability to promote gene conversions and suppress SDDs depends on its ATPase and associated DNA helicase/translocase activity, we next tested the effect of mutating conserved residues in its Walker A (K99R) and Walker B (D196N) motifs, which render Fml1 catalytically inactive without affecting its ability to bind DNA (*Nandi and Whitby, 2012*). Both mutants exhibit a similar reduction in gene conversions and

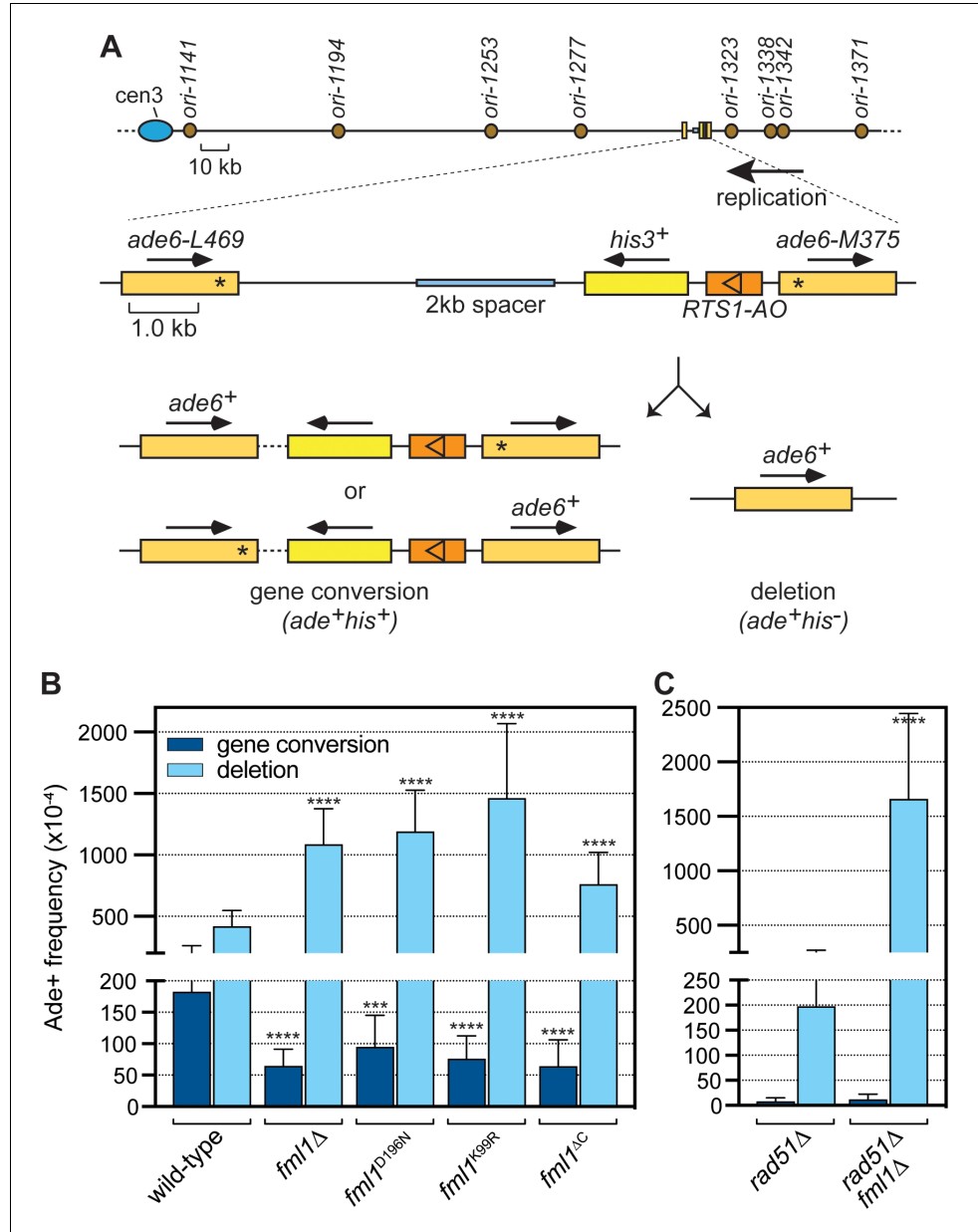

**Figure 1.** Comparison of SDD and gene conversion frequency in different *fml1* mutant strains. (**A**) Schematic of the direct repeat recombination reporter and its position on chromosome 3. Key features, including replication origins, prevailing direction of replication at the reporter, 2 kb spacer DNA and different types of Ade[+] recombinant are indicated. Asterisks mark the position of point mutations in *ade6-L469* and *ade6-M375*. (**B**) Ade[+] recombinant frequencies for strains MCW8020 (wild-type), MCW8300 (*fml1Δ*), MCW9268 (*fml1*[D196N]), MCW9281 (*fml1*[K99R]), and MCW9266 (*fml1*[ΔC]). (**C**) Ade[+] recombinant frequencies for strains MCW8296 (*rad51Δ*) and MCW9496 (*rad51Δ fml1Δ*). Data are mean values with error bars showing 1 SD. Significant differences relative to equivalent wild-type values are indicated by *p<0.05, **p<0.01, ***p<0.001, ****p<0.0001. Ade[+] recombinant frequencies with statistical analysis are also shown in **Supplementary file 1**.

increase in SDDs as a *fml1Δ* mutant, indicating that Fml1's catalytic activity is essential for its suppression of IFSA (**Figure 1B**). We next tested whether Fml1's C-terminal domain is required for its suppression of IFSA using a *fml1*[ΔC] mutant, which encodes a truncated form of Fml1 missing its terminal 231 amino acids that nevertheless retains DNA binding and key catalytic activities (**Sun et al., 2008**) (**Figure 1B**). This *fml1*[ΔC] mutant exhibited a similar reduction in gene conversions as a *fml1Δ* mutant (p=0.961). However, whilst it also displayed a significant increase in SDDs compared to wild-

type, the increase was noticeably less than observed in a *fml1Δ* mutant (p<0.001). These data indicate that Fml1's C-terminal domain is required for its ability to promote gene conversions but only partially required for its ability to suppress SDDs.

## Fml1 suppresses Rad51-independent SDDs

We showed previously that *RTS1*-induced SDDs can form without Rad51 in a Rad52-dependent process (*Morrow et al., 2017*). Indeed, with a 5 kb DNA spacer the frequency of SDDs in wild-type and *rad51Δ* mutant is essentially the same (*Morrow et al., 2017*). However, we had not determined whether Fml1 suppresses Rad51-independent SDDs and, therefore, we compared the frequency of recombination in a *rad51Δ* single mutant with that in a *rad51Δ fml1Δ* double mutant using the recombination reporter with a 2 kb spacer (*Figure 1A,C*). Consistent with Rad51's known role in promoting gene conversions, both mutants exhibit only residual levels of gene conversions compared to the equivalent *rad51*⁺ strains (*Figure 1B,C*). In a *rad51Δ* single mutant, the frequency of SDDs is ~2 fold less than in wild-type (p<0.0001) indicating that some deletions are formed by Rad51. However, in a *rad51Δ fml1Δ* double mutant SDDs increase by ~4 fold compared to wild-type (p<0.0001) and ~8 fold compared to a *rad51Δ* single mutant (p<0.0001) (*Figure 1B,C*). These data show that Fml1 plays an important role in supressing Rad51-independent SDDs.

## Fml1 interacts with Mhf1-Mhf2 via its C-terminal domain

Our lab previously reported that the C-terminal domain of Fml1 interacts with Mhf1-Mhf2 (MHF) (*Bhattacharjee et al., 2013*). It was also reported that mutation of three amino acids (Y672, R674 and R678) to alanine within this domain (henceforth referred to as the 'AAA' mutation) disrupts Fml1's interaction with MHF (*Bhattacharjee et al., 2013*). To reaffirm these findings, we tested different fragments of Fml1, which were fused to maltose binding protein (MBP) and bound to amylose resin, for their ability to 'pull-down' purified MHF (*Figure 2A,B*). Under low salt conditions (10 mM NaCl), several different C-terminal fragments of Fml1, encompassing amino acids 576–690, including one containing the AAA mutation, were able to pull-down MHF, whereas two N-terminal fragments (encompassing amino acids 1–575 and 1–603) and a MBP control could not (*Figure 2C*). However, under high salt conditions (600 mM NaCl), robust pull-down was only achieved with a Fml1$^{576-834}$ fragment, with shorter fragments (Fml1$^{576-690}$ and Fml1$^{576-725}$) and Fml1$^{401-730}$ showing reduced levels of MHF pull-down (*Figure 2D*). Moreover, consistent with our previous findings, AAA mutation of Fml1$^{576-834}$ strongly reduces its ability to pull-down MHF under high salt conditions (*Figure 2D*) (*Bhattacharjee et al., 2013*). However, despite reducing the interaction between Fml1$^{576-834}$ and MHF in vitro, the AAA mutation (*fml1$^{AAA}$*) had little or no effect on resistance to the genotoxins methyl methanesulfonate (MMS), ultra-violet light (UV) and cisplatin in vivo, whereas a *fml1$^{ΔC}$* mutant exhibited even greater hypersensitivity to these agents than a *fml1Δ* mutant (*Figure 2—figure supplement 1*). These data suggest that either the AAA mutation does not sufficiently disrupt Fml1's interaction with MHF in vivo to affect its DNA repair role or that Fml1's C-terminal domain is important for some other reason than interacting with MHF. It should be noted that the original phenotypes reported for *fml1$^{AAA}$* and *fml1$^{ΔC}$* mutants were found to be incorrect due to problems with the strains that were tested (*Bhattacharjee et al., 2013*; *Bhattacharjee et al., 2018*).

To further explore the topology of the interaction between Fml1 and MHF, we co-expressed full-length Fml1 and MHF in *Escherichia coli* and purified them as a complex by nickel affinity and gel filtration chromatography (*Figure 2E*). The purified complex was then subjected to lysine-specific crosslinking followed by tandem mass spectrometry analysis (*Leitner et al., 2014*) to identify the relative spatial relationship of Fml1's N- and C-terminal domains with MHF (*Figure 2F*). This analysis revealed extensive crosslinking between lysines in Mhf1 and Mhf2 with lysines throughout Fml1's C-terminal domain. However, significant crosslinking within much of Fml1's N-terminal helicase domain was not detected. Altogether the crosslinking and pull-down data suggest that Fml1 interacts with MHF by an extended region of contacts across its C-terminal domain, which is consistent with data from a co-crystal structure of human MHF1, MHF2 and FANCM$^{661-800}$ (*Tao et al., 2012*). Presumably, the presence of multiple contact points explains why the AAA mutation is insufficient to fully disrupt the interaction between Fml1 and MHF.

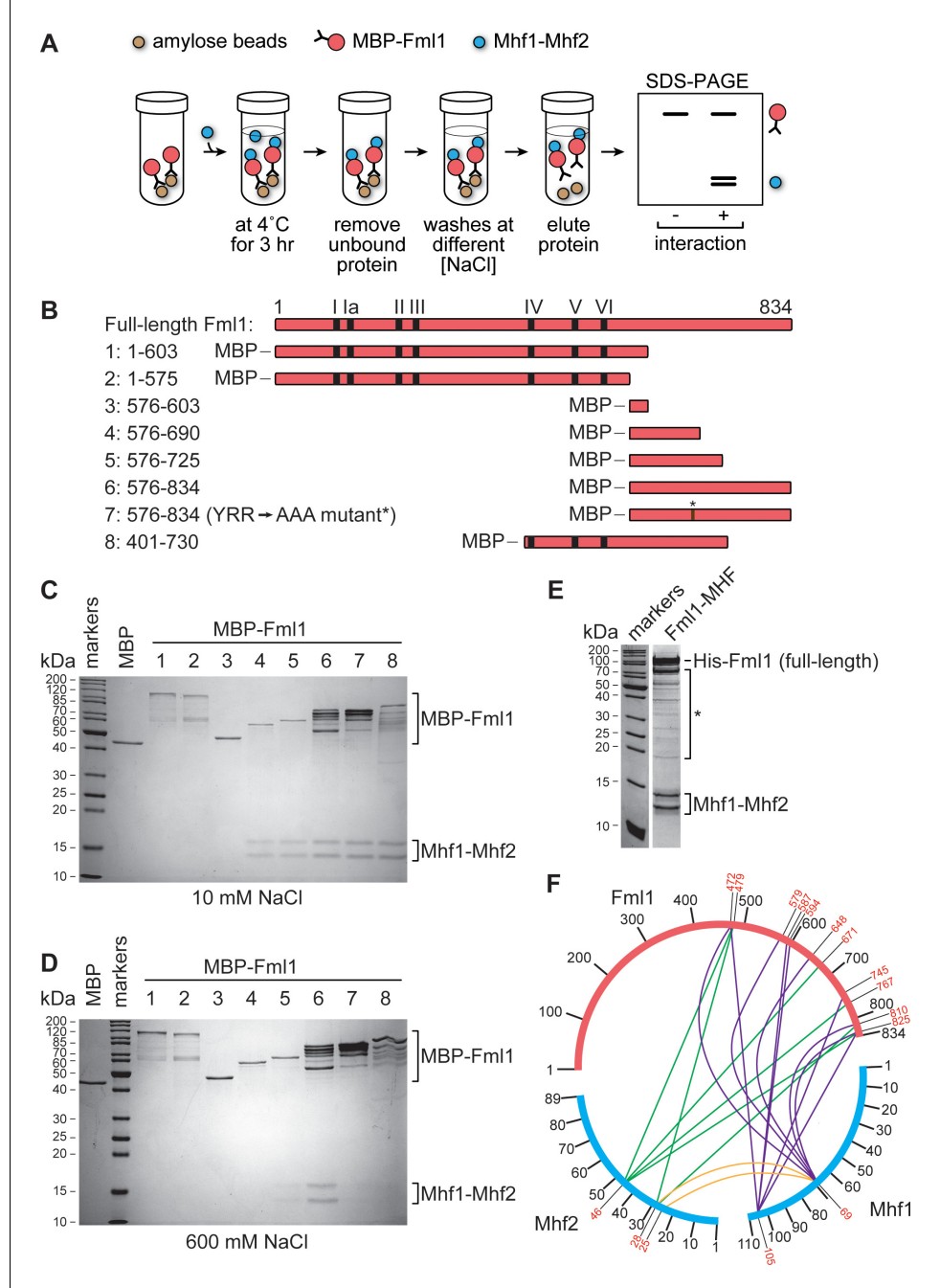

**Figure 2.** Mapping the region of Fml1 that interacts with Mhf1-Mhf2 (MHF). (A) Experimental scheme for investigating the interaction between different MBP fused fragments of Fml1 and purified MHF. (B) Schematic of eight different Fml1 fragments tested for interaction with MHF. The terminal amino acids of each fragment are indicated by the numbers on the left-hand side of the panel. The position of the seven conserved helicase motifs (I – VI) in Fml1 and YRR to AAA mutation (*) are indicated. (C) Coomassie blue stained SDS-PAGE gel showing the results of pull-down experiments, with 10 mM NaCl washes, using the eight different MBP fused Fml1 fragments shown in B plus an MBP control. (D) The same as C except the washes were 600 mM NaCl. (E) Coomassie blue stained SDS-PAGE gel showing purified Fml1-MHF complex (4.3 μg). Truncated fragments of Fml1 and trace contaminant proteins are indicated by the asterisk. (F) Circos-like plot depicting crosslinking mass spectrometry analysis of the Fml1-MHF complex. Crosslinked lysines with a confidence score cut-off >15 are numbered in red. The online version of this article includes the following figure supplement(s) for figure 2:

*Figure 2 continued on next page*

*Figure 2 continued*

**Figure supplement 1.** Spot assay comparing the genotoxin sensitivities of strains MCW1221 (wild-type), MCW2080 (*fml1Δ*), MCW6923 (*fml1ΔC*) and MCW6976 (*fml1AAA*).

## Identification of a mutation in Mhf2 that causes MMS hypersensitivity but does not disrupt MHF's centromeric function

To further investigate whether the key purpose of Fml1's C-terminal domain is to mediate its interaction with MHF, we screened for additional mutations within the complex that would disrupt the interaction. In particular we looked for mutations within Mhf2 that would specifically affect its role with Fml1 without affecting its centromeric function. Unlike a *fml1Δ* mutant, a *mhf2Δ* mutant exhibits very poor growth that is symptomatic of its key centromeric function (*Figure 3A*) (*Bhattacharjee et al., 2013*). We reasoned that the separation of function mutation in Mhf2 that we were looking for would cause hypersensitivity to MMS without affecting growth in the absence of genotoxin. Guided by a crystal structure of *S. pombe* MHF (unpublished data), we tested alanine mutations in three residues (S59, Q83 and D87) that we thought might interfere with the interaction with Fml1 (*Figure 3A*). One of these, D87A, exhibited the phenotype we were looking for. To confirm that the D87A mutation was specifically affecting MHF's function with Fml1, we compared the MMS sensitivities of *fml1Δ* and *mhf2*$^{D87A}$ single and double mutants (*Figure 3B*). This analysis revealed an epistatic genetic interaction, with the double mutant exhibiting the same sensitivity as a *fml1Δ* single mutant, which is consistent with the D87A mutation only affecting MHF's Fml1-related function. We also analysed chromosome segregation in the *mhf2*$^{D87A}$ mutant and found that less than 10% of septated cells exhibited failed or aberrant chromosome segregation (similar to a *fml1Δ* mutant), compared to >40% in a *mhf2Δ* mutant (*Figure 3—figure supplement 1*). These data provide further evidence that the D87A mutation is specific to MHF's Fml1-related function and has little or no effect on its ability to perform its centromeric role.

To investigate how the D87A mutation disrupts MHF function, we first tested whether its DNA binding activity was perturbed in vitro (*Figure 3—figure supplement 2*). The Mhf1 plus (His)$_6$ tagged Mhf2$^{D87A}$ complex was purified by nickel affinity, gel filtration and heparin affinity chromatography. Like Mhf1-Mhf2, Mhf1-Mhf2$^{D87A}$ purifies as a 1:1 complex, although this is not apparent by SDS-PAGE analysis as (His)$_6$ tagged Mhf2$^{D87A}$ exhibits faster migration than its non-mutant form and consequently co-migrates with Mhf1 (*Figure 3—figure supplement 2A*). Human MHF exhibits weak DNA binding to linear dsDNA and various branched DNA substrates, including synthetic Holliday junctions (*Singh et al., 2010*; *Yan et al., 2010*). *S. pombe* MHF exhibits even weaker DNA binding in vitro than its human counterpart but, importantly, this is not affected by the D87A mutation, as judged by its binding to a synthetic Holliday junction (X-12) in an electrophoretic mobility shift assay (EMSA) (*Figure 3—figure supplement 2B*).

We next tested whether the D87A mutation affected the level of MHF in the cell. It was previously shown that MHF1 and MHF2 depend on each other for stability in HeLa cells (*Yan et al., 2010*). Therefore, if the D87A mutation causes a reduction in the level of Mhf2 then we should see a corresponding reduction in the level of Mhf1. To monitor Mhf1 levels, we used a strain in which Mhf1 is tagged with green fluorescent protein (GFP) and expressed from its native promoter at its endogenous locus (*Bhattacharjee et al., 2013*). Consistent with the finding in HeLa cells, deletion of *mhf2* results in a loss of Mhf1-GFP stability (*Figure 3—figure supplement 3A*). In contrast, neither D87A mutation nor *fml1* deletion had a noticeable effect on Mhf1-GFP stability (*Figure 3—figure supplement 3A*). From these data we infer that the level of Mhf2/MHF in the cell is not affected by the D87A mutation or absence of Fml1.

We next examined whether MHF's cellular localization is affected. Consistent with previous work from our lab, we observed Mhf1-GFP co-localizing with the centromeric protein Mis6 (CENP-I), tagged with mCherry (*Figure 3—figure supplement 3B*) (*Bhattacharjee et al., 2013*). This centromeric localization of Mhf1-GFP is not disrupted by the D87A mutation in Mhf2 or by deletion of *fml1* (*Figure 3—figure supplement 3B*). As expected, no centromeric Mhf1-GFP was detected in a *mhf2Δ* mutant (*Figure 3—figure supplement 3B*). In addition to localizing to centromeres, Mhf1-GFP also exhibits a diffuse pattern of fluorescence throughout the nucleus, which depends on Fml1 (*Figure 3—figure supplement 3B*) (*Bhattacharjee et al., 2013*). This non-centromeric localization of

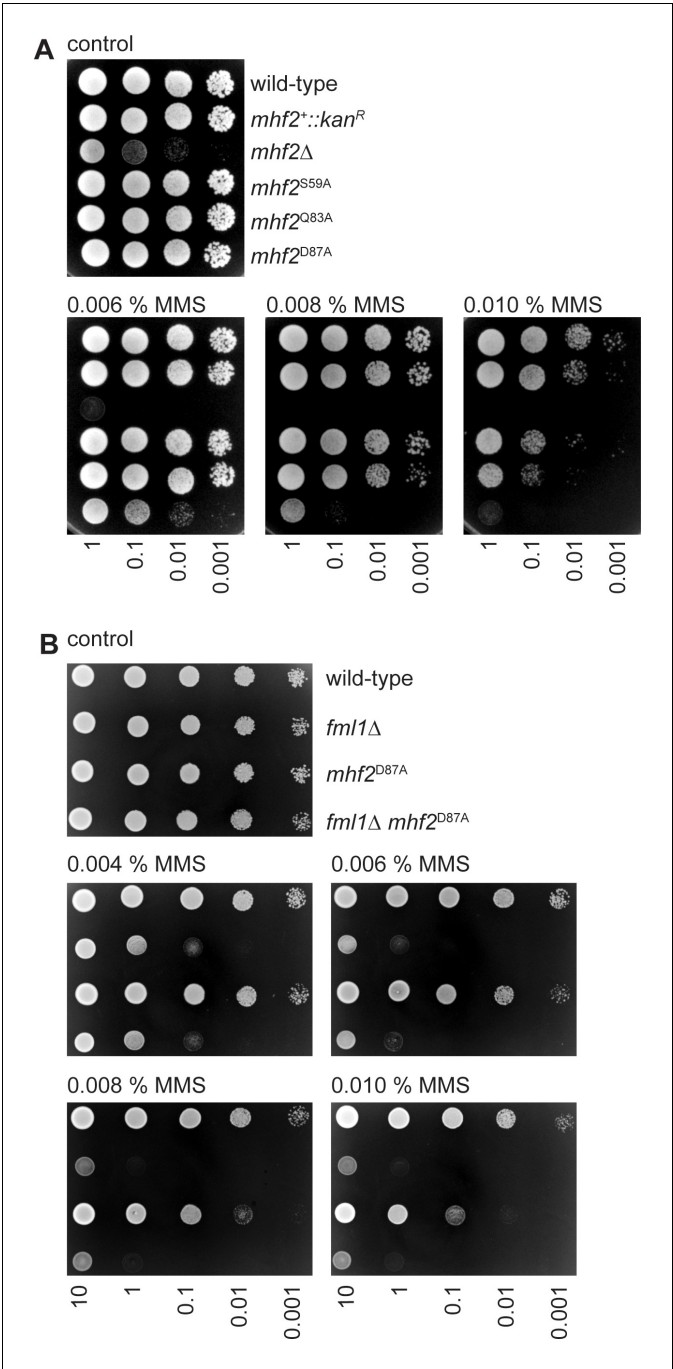

**Figure 3.** Spot assays comparing the MMS sensitivities of indicated *mhf2* and *fml1* mutant strains. (**A**) The strains are MCW1221 (wild-type), MCW5677 (*mhf2+::kan^R*), MCW5112 (*mhf2Δ*), MCW6376 (*mhf2^S59A*), MCW6379 (*mhf2^Q83A*) and MCW6367 (*mhf2^D87A*). (**B**) The strains are FO808 (wild-type), MCW4708 (*fml1Δ*), MCW6319 (*mhf2^D87A*) and MCW6570 (*fml1Δ mhf2^D87A*). Note that the apparent difference in MMS sensitivity of the *mhf2^D87A* strain in the experiments shown in panel A and B reflects differences in the potency of MMS between different batches of plates.

The online version of this article includes the following figure supplement(s) for figure 3:

**Figure supplement 1.** Analysis of chromosome segregation in septated cells.
**Figure supplement 2.** Comparison of DNA binding by Mhf1-Mhf2 and Mhf1-Mhf2^D87A.
**Figure supplement 3.** Analysis of Mhf1-GFP in wild-type, *mhf2Δ*, *mhf2^D87A* and *fml1Δ* cells.

Mhf1-GFP is lost or greatly diminished in a $mhf2^{D87A}$ mutant. Together these data suggest that the D87A mutation in Mhf2 disrupts MHF's ability to localize to non-centromeric DNA and, in this regard, it phenocopies a $fml1\Delta$ mutant.

## A D87A mutation in Mhf2 disrupts the interaction between Fml1 and MHF

Both the MMS hypersensitivity and reduced Mhf1-GFP non-centromeric nuclear fluorescence of a $mhf2^{D87A}$ mutant could be explained by an impairment of the interaction between Fml1 and MHF. To directly test this, we compared the pull-down of Mhf1-Mhf2 and Mhf1-Mhf2$^{D87A}$ by MBP-Fml1$^{604-834}$ at different salt concentrations using the assay outlined in *Figure 2A* (*Figure 4A,B*). Similar to MBP-Fml1$^{576-834}$, MBP-Fml1$^{604-834}$ efficiently pulled down Mhf1-Mhf2 over a range of salt concentrations (50–600 mM NaCl) (*Figure 4A*, lanes c - f). In contrast, the pull-down of Mhf1-Mhf2$^{D87A}$, whilst efficient at low salt (50 mM NaCl), was greatly diminished at high salt (600 mM NaCl) (*Figure 4B*, lanes c and f). These data indicate that the D87A mutation in Mhf2 impairs the interaction between MHF and Fml1's C-terminal domain.

To confirm that the D87A mutation also impairs the interaction between Fml1 and MHF in vivo, we compared the ability of 13Myc-tagged Fml1 to co-immunopreciptate (co-IP) with Mhf1-GFP from $mhf2^{+}$ and $mhf2^{D87A}$ strains (*Figure 4—figure supplement 1*). Even though the amount of immuno-preciptated Fml1-13Myc was less from $mhf2^{D87A}$ mutant cells than from $mhf2^{+}$ cells, it was still evident that Fml1's interaction with Mhf1 is impaired by the D87A mutation in Mhf2.

## Combining Fml1$^{AAA}$ and Mhf2$^{D87A}$ mutants greatly impairs Fml1$^{604-834}$-MHF complex formation in vitro and causes a synergistic increase in MMS sensitivity in vivo

Having established that both Fml1$^{AAA}$ and Mhf2$^{D87A}$ mutants impair Fml1-MHF complex formation under high salt conditions, we wondered whether the combination of both mutants would lead to an even greater impairment. As seen previously with the AAA mutant of MBP-Fml1$^{576-834}$, MBP-Fml1$^{604-834AAA}$ was able to pull-down Mhf1-Mhf2 under low salt conditions (50 mM NaCl) but not at high salt (600 mM NaCl) (*Figure 4A*, lanes g and j). The same is true for the pull-down of Mhf1-Mhf2$^{D87A}$ by MBP-Fml1$^{604-834AAA}$ (*Figure 4B*, lanes g and j). However, a clear difference is observed at intermediate salt concentrations (200–300 mM NaCl), with MBP-Fml1$^{604-834AAA}$ being able to pull-down Mhf1-Mhf2 but not Mhf1-Mhf2$^{D87A}$ under these conditions (*Figure 4A,B*, lanes h and i). These data indicate that the combination of Fml1$^{AAA}$ and Mhf2$^{D87A}$ mutants has at least an additive effect on weakening Fml1-MHF complex formation in vitro. However, this effect is not apparent in our co-IP experiment as, under these conditions, the D87 mutation in Mhf2 is already sufficient to cause loss of detectable levels of Mhf1-GFP in immunoprecipitates of Fml1-13Myc regardless of whether Fml1 is carrying the AAA mutation or not (*Figure 4—figure supplement 1*). Nevertheless, the reduction in complex formation detected in vitro correlates with a synergistic increase in MMS sensitivity of a $fml1^{AAA}$ $mhf2^{D87A}$ double mutant in vivo, albeit the double mutant is still not quite as sensitive as a $fml1\Delta$ or $fml1^{\Delta C}$ mutant (*Figure 4C*). Altogether these data are consistent with the notion that a key function of Fml1's C-terminal domain is to mediate its interaction with MHF, which in turn is important for its ability to promote DNA repair.

## A $fml1^{AAA}mhf2^{D87A}$ double mutant exhibits reduced gene conversions and increased SDDs

We next evaluated the effect of $fml1^{AAA}$ and $mhf2^{D87A}$ mutations on the frequency of $RTS1$-induced gene conversions and SDDs (*Figures 1A* and *5*). Both single mutants exhibited an increase in SDDs, compared to wild-type, which in the $fml1^{AAA}$ was similar to a $fml1^{\Delta C}$ mutant (p=0.38), but for $mhf2^{D87A}$ was slightly less (p<0.05). In contrast, gene conversions were either not reduced ($fml1^{AAA}$) or only reduced 1.5-fold ($mhf2^{D87A}$) relative to wild-type, which is less than the 3-fold reduction seen in a $fml1^{\Delta C}$ mutant (p<0.01). Unlike the single mutants, the $fml1^{AAA}$ $mhf2^{D87A}$ double mutant more clearly recapitulated the phenotype of a $fml1^{\Delta C}$ mutant with regards to both SDD and gene conversion frequency, albeit gene conversion frequency was still slightly higher than in the $fml1^{\Delta C}$ mutant (p<0.05). Altogether these data suggest that the relatively modest disruption of Fml1-MHF complex formation, seen with Fml1$^{AAA}$ and Mhf2$^{D87A}$ mutant proteins, is already sufficient to cause a defect

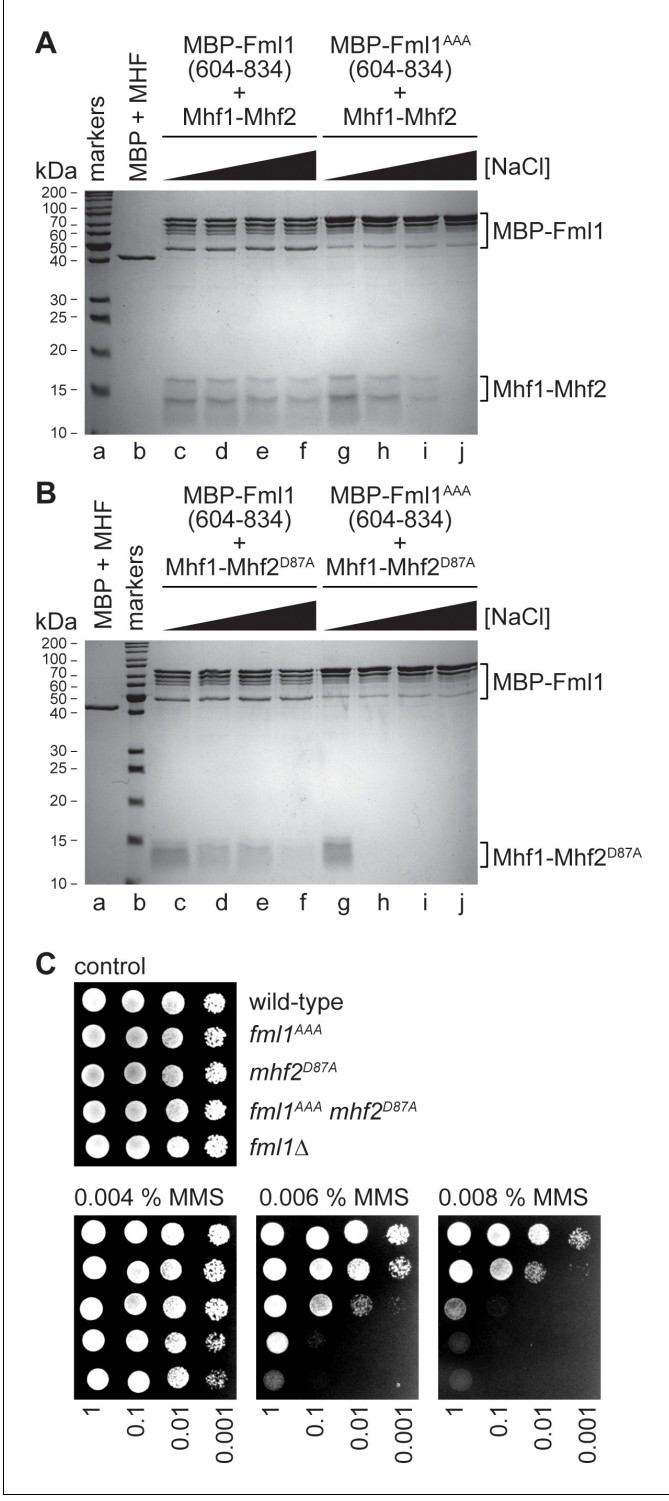

**Figure 4.** Effect of the AAA mutation in Fml1 and D87A mutation in Mhf2 on the interaction between Fml1 and MHF. (**A**) Coomassie blue stained SDS-PAGE gel showing the results of pull-down experiments (see *Figure 2A*) using MBP-Fml1$^{604-834}$/MBP-Fml1$^{604-834AAA}$ and Mhf1-Mhf2 performed with increasing NaCl concentration washes (50 mM, 200 mM, 300 mM and 600 mM). The result of a control pull-down experiment using purified MBP and Mhf1-Mhf2 at 10 mM NaCl is also included. (**B**) The same as in **A**) except Mhf1-Mhf2$^{D87A}$ was used instead of Mhf1-Mhf2. (**C**) Spot assay comparing the MMS sensitivities of strains MCW1221 (wild-type), MCW6976 (*fml1*$^{AAA}$), MCW6321 (*mhf2*$^{D87A}$), MCW7057 (*fml1*$^{AAA}$ *mhf2*$^{D87A}$) and MCW2080 (*fml1Δ*).

*Figure 4 continued on next page*

*Figure 4 continued*

The online version of this article includes the following figure supplement(s) for figure 4:

**Figure supplement 1.** Co-immunoprecipitation of Fml1$^+$–13Myc/Fml1$^{AAA}$-13Myc and Mhf1-GFP from *mhf2$^+$* and *mhf2$^{D87A}$* cells.

in SDD suppression, whereas Fml1's role in promoting gene conversions seems to be a bit more tolerant of this level of disruption, with a combination of both AAA and D87A mutations being needed to most closely resemble the defect in a *fml1$^{\Delta C}$* mutant.

## A D87A mutation in Mhf2 destabilizes Fml1 in vivo

Depletion of either MHF1 or MHF2 in HeLa cells causes a reduction in the stability of FANCM (*Yan et al., 2010*). To see whether interaction with MHF is similarly required for Fml1's stability in *S. pombe*, we first assessed the levels of Fml1-13Myc, Fml1$^{\Delta C}$-13Myc and Fml1$^{AAA}$-13Myc in whole cell extracts (*Figure 6A*). This analysis showed that there was little or no difference in the levels of Fml1-13Myc and Fml1$^{\Delta C}$-13Myc, whereas there was a noticeable increase in the amount of Fml1$^{AAA}$-13Myc. We next assessed whether the D87A mutation in Mhf2 affects Fml1 stability (*Figure 6B*). In this case, we observed a marked reduction in the level of Fml1-13Myc in the *mhf2$^{D87A}$* mutant indicating that the D87A mutation affects Fml1 stability. Interestingly, despite the AAA mutation weakening interaction with MHF in vitro, Fml1$^{AAA}$-13Myc exhibited better stability in a *mhf2$^{D87A}$* mutant than Fml1-13Myc (*Figure 6B*). Altogether these data suggest that Fml1 is rendered unstable by its C-terminal domain and this instability is suppressed by interaction with MHF. Interestingly the AAA mutation helps to stabilise Fml1 both in the presence and absence of MHF interaction. Based on this finding, we can also conclude that the heightened phenotypes of a *fml1$^{AAA}$ mhf2$^{D87A}$* double mutant compared to a *mhf2$^{D87A}$* single mutant are not due to further reduction in the cellular levels of Fml1. Instead, the observed differences in recombination and DNA repair between double and single mutant likely reflect the fact that MHF's interaction with Fml1 plays an additional role beyond simply promoting protein stability.

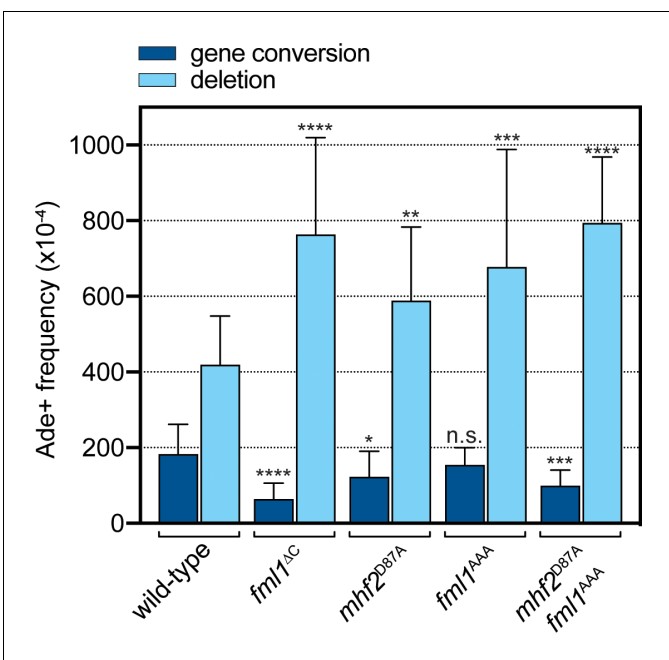

**Figure 5.** Effect of the AAA mutation in Fml1 and D87A mutation in Mhf2 on *RTS1*-induced recombination. Comparison of SDD and gene conversion frequency in strains MCW8020 (wild-type), MCW9266 (*fml1$^{\Delta C}$*), MCW9220 (*mhf2$^{D87A}$*), MCW9283 (*fml1$^{AAA}$*) and MCW9269 (*fml1$^{AAA}$ mhf2$^{D87A}$*). Ade$^+$ recombinant frequencies with statistical analysis are also shown in *Supplementary file 1*.

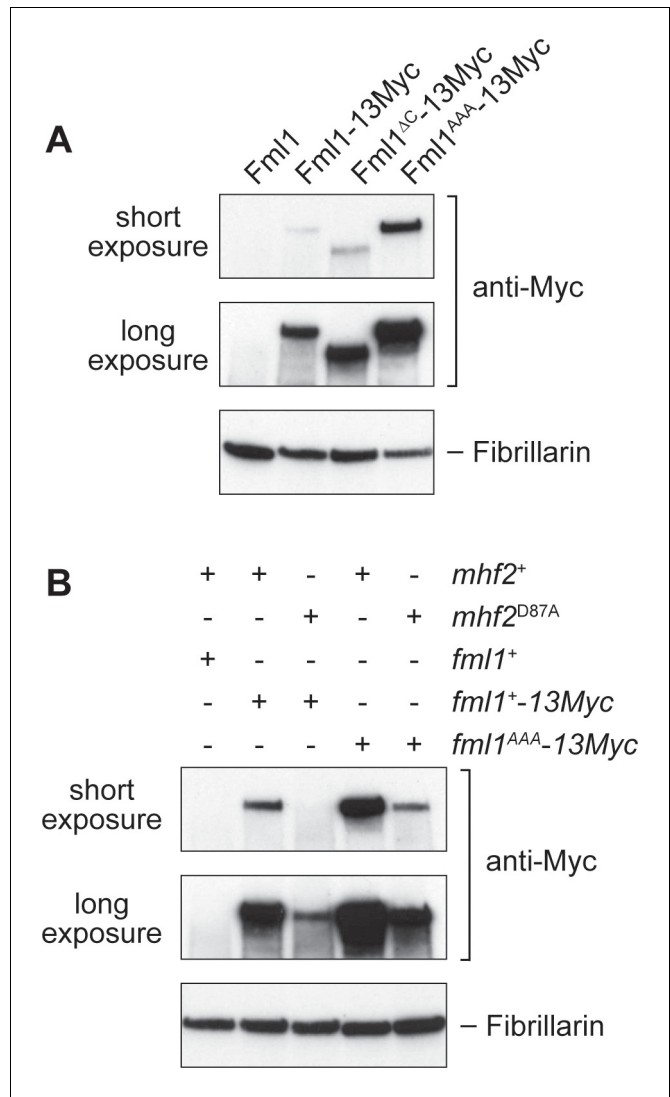

**Figure 6.** Effect of the AAA mutation in Fml1 and D87A mutation in Mhf2 on Fml1 stability. (**A**) Western blot showing the relative amounts of Fml1$^+$−13Myc, Fml1$^{\Delta C}$-13Myc and Fml1$^{AAA}$-13Myc in whole cell extracts from strains MCW1221 (*fml1$^+$*), MCW4406 (*fml1$^+$−13Myc*), MCW6977 (*fml1$^{\Delta C}$-13Myc*), and MCW6980 (*fml1$^{AAA}$-13Myc*). (**B**) Western blot showing the relative amounts of Fml1$^+$−13Myc/Fml1$^{AAA}$-13Myc in whole cell lysates from strains MCW1221 (*fml1$^+$ mhf2$^+$*), MCW9616 (*fml1$^+$−13Myc mhf2$^+$*), MCW9593 (*fml1$^+$−13Myc mhf2$^{D87A}$*), MCW9594 (*fml1$^{AAA}$-13Myc mhf2$^+$*) and MCW9595 (*fml1$^{AAA}$-13Myc mhf2$^{D87A}$*). Fibrillarin serves as a loading control.

## Fml1$^{\Delta C}$ and Fml1-MHF catalyse replication fork restoration in vitro

Morrow et al suggested two ways that Fml1 could suppress SDDs: 1) it could catalyse fork restoration, which would limit the size of the 3' ssDNA tail at the regressed fork and in so-doing reduce the opportunity for Rad52-mediated IFSA (*Figure 7*, steps 2a, 3a and 4a); and 2) it could unwind the putative 'IFSA junction' that is created when Rad52 anneals the 3' ssDNA tail at the regressed fork into the lagging strand gap of the oncoming replication fork (*Figure 7*, step 5a) (*Morrow et al., 2017*). They also suggested that fork restoration might drive unwinding of the IFSA junction (*Figure 7*, step 5b). To investigate whether Fml1 is capable of catalysing fork restoration, we used a set of synthetic regressed replication fork substrates that were originally developed to test the ability of human SMARCAL1 to catalyse fork restoration (*Figure 8A*) (*Bétous et al., 2013*). Two of these substrates have either a 3' or 5' 32 nucleotide (nt) ssDNA tail (referred to as lagging gap and leading gap forks, respectively), whereas the third has a 32 bp duplex tail (mimicking a regressed fork that has not been resected). The substrates contain heterologous DNA arms, which restricts the reaction

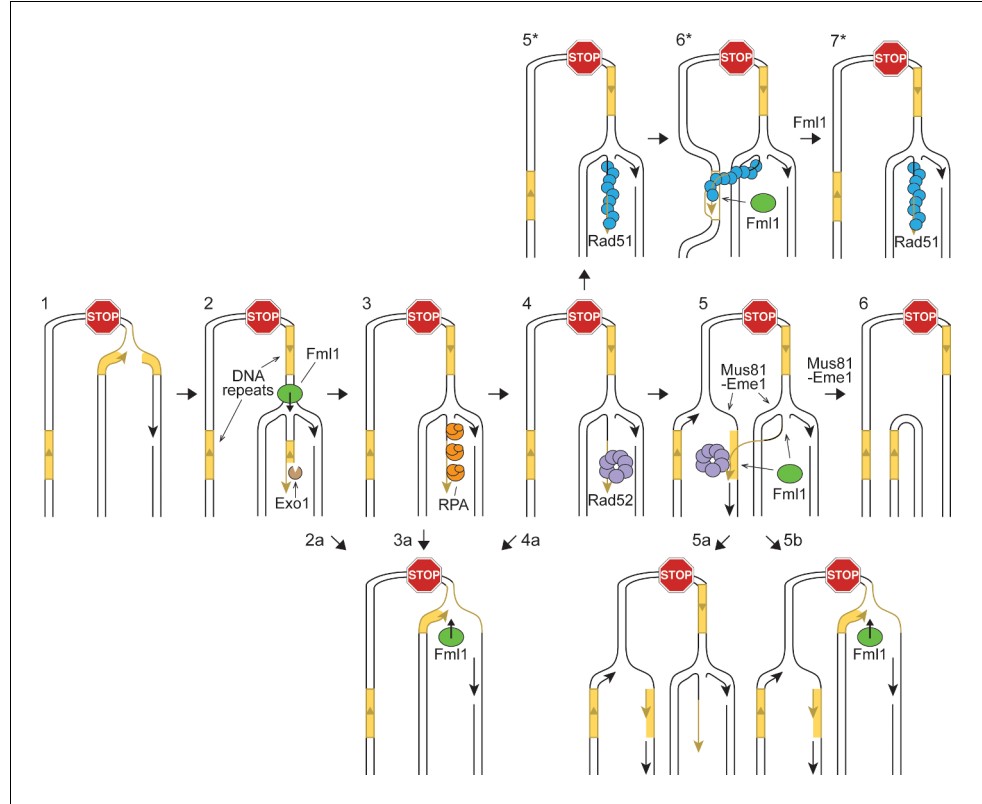

**Figure 7.** Hypothetical model showing potential ways in which Fml1 could promote gene conversions and suppress SDDs formed by recombination between DNA repeats at a collapsed replication fork. Step 1: Replication fork collapses at RFB. Step 2: Replication fork regresses (Fml1 may help to catalyse this process) and Exo1 resects the exposed dsDNA end. Step 3: RPA binds to the ssDNA tail at the resected regressed fork. Step 4: Rad52 binds to the ssDNA tail at the resected regressed fork. Step 5: Rad52 anneals the ssDNA at the regressed fork to the lagging strand gap of the oncoming replication fork to form an 'IFSA junction'. Step 6: Mus81-Eme1 cleaves the IFSA junction generating a SDD. Steps 2a-4a: Fml1 catalyses fork restoration limiting Rad52-mediated IFSA. Step 5a: Fml1 unwinds the IFSA junction. Alternatively, it may catalyse fork restoration like in Steps 2a-4a and in so-doing disrupt the IFSA junction (Step 5b). Step 5*: Rad52 mediates the loading of Rad51 onto the ssDNA tail at the regressed fork. Step 6*: Rad51 catalyses strand invasion of the second DNA repeat leading to the formation of a D-loop at which DNA synthesis is primed. Following limited extension of the invading strand by a DNA polymerase (not shown), the D-loop is unwound by Fml1 (Steps 6* - 7*). If the DNA copied during the strand invasion event is not identical in sequence to the first repeat, then fork restoration will lead to the formation of a heteroduplex DNA with the potential for gene conversion either by mismatch repair or by DNA replication (not shown).

to fork restoration only, and base pair mismatches that prevent spontaneous branch migration. We first tested whether Fml1$^{\Delta C}$ could catalyse fork restoration by incubating increasing concentrations of the protein with the three different substrates (*Figure 8B,C*). This established that Fml1$^{\Delta C}$ has a clear preference for restoring a lagging gap fork as ~20 fold less protein was required to achieve the same level of fork restoration within 20 min with this substrate compared to the leading gap and regressed fork substrates (*Figure 8C*). We also confirmed that fork restoration depends on Fml1's ATPase activity, as Fml1$^{\Delta CK99R}$ was unable to catalyse this reaction with any of the three substrates (*Figure 8—figure supplement 1*). Next, we tested the ability of the Fml1-MHF complex to catalyse fork restoration (*Figure 8D,E*). Similar to Fml1$^{\Delta C}$, the Fml1-MHF complex exhibits a preference for restoration of the lagging gap fork, albeit it performs less well than Fml1$^{\Delta C}$ at sub-stoichiometric concentrations (i.e. at a ratio of 0.1 nM protein to 0.5 nM lagging gap substrate) (*Figure 8E*). It is also slightly better at restoring the regressed fork substrate than Fml1$^{\Delta C}$ (*Figure 8E*). To further evaluate the ability of Fml1$^{\Delta C}$ and Fml1-MHF to catalyse fork restoration, we performed time-course

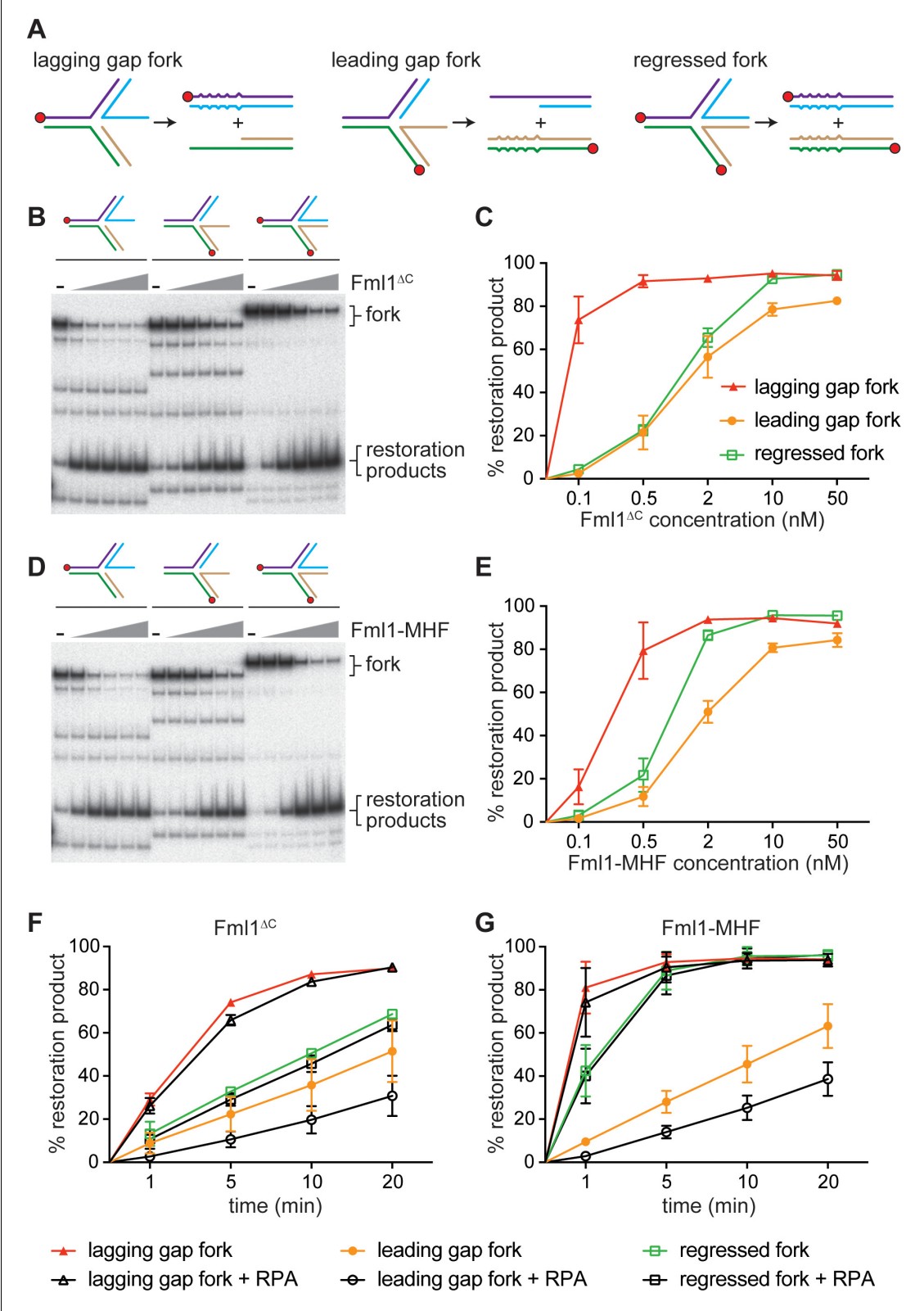

**Figure 8.** Replication fork restoration by Fml1$^{\Delta C}$ and Fml1-MHF. (**A**) Schematic of the regressed fork substrates and reaction products. Component oligonucleotides are colour coded and red circles indicate 5' $^{32}$P labels. Base pair mismatches in the reaction products are indicated by zigzagged lines. (**B**) Comparison of the restoration of the different regressed fork substrates shown in (**A**) by increasing concentrations of Fml1$^{\Delta C}$. (**C**) Quantification of data like in B. (**D**) The same as (**B**) but with Fml1-MHF instead of Fml1$^{\Delta C}$. (**E**) Quantification of data like in C. (**F** – **G**) Time course reactions for the

*Figure 8 continued on next page*

*Figure 8 continued*

restoration of the different regressed fork substrates shown in **A** by Fml1$^{\Delta C}$ (0.5 nM with lagging gap fork and 2 nM with leading gap fork and regressed fork) and Fml1-MHF (0.5 nM with lagging gap fork and 2 nM with leading gap fork and regressed fork) in the presence and absence of RPA (20 nM).

Data in (**C**) and (**E – G**) are mean values +/- SD from three independent experiments.

The online version of this article includes the following figure supplement(s) for figure 8:

**Figure supplement 1.** Restoration of regressed fork substrates by Fml1 requires its ATPase activity.

**Figure supplement 2.** Effect of Rad52 on lagging gap fork restoration by Fml1-MHF.

experiments using 0.5 nM of DNA substrate and either 0.5 nM protein (in the case of the lagging gap fork) or 2 nM protein (in the case of the leading gap and regressed forks) (*Figure 8F,G*). Even with less protein added, the lagging gap fork was restored, by both Fml1$^{\Delta C}$ and Fml1-MHF, with faster kinetics than either the leading gap or regressed forks. Intriguingly, Fml1-MHF exhibited noticeably faster restoration kinetics than Fml1$^{\Delta C}$ on both the lagging gap and regressed fork substrates, even though Fml1$^{\Delta C}$ is better at restoring the lagging gap fork at sub-stoichiometric concentrations (*Figure 8C,E,F,G*). A possible explanation for these seemingly conflicting observations is given in the Discussion.

In vivo, the ssDNA tail at a resected regressed replication fork would be bound first by the ssDNA binding protein RPA, followed by Rad52 (*Figure 7*, steps 3 and 4). The 32 nt ssDNA tail on both the lagging and leading gap forks is designed to accommodate one molecule of RPA in its high-affinity DNA-bound state (*Bétous et al., 2013*) and, therefore, we investigated whether the pre-incubation of these substrates with RPA affected the time-course of fork restoration by either Fml1$^{\Delta C}$ or Fml1-MHF (*Figure 8F,G*). RPA had little or no effect on the rate of restoration of either lagging gap or regressed forks by either Fml1$^{\Delta C}$ or Fml1-MHF. However, it reduced the rate of leading gap fork restoration by ~2 fold with both proteins.

We next investigated whether Rad52 affects Fml1-MHF's ability to restore a lagging gap fork (*Figure 8—figure supplement 2*). A 60-fold molar excess of Rad52 (assuming a heptameric conformation of Rad52) was incubated with the lagging gap fork substrate. Increasing concentrations of Fml1-MHF (ranging from a 2- to 20-fold molar excess over DNA substrate) were then added and incubation continued at 37 °C for 20 min (*Figure 8—figure supplement 2A*). The presence of Rad52 had a marked inhibitory effect on fork restoration at the concentrations of Fml1-MHF tested, with the amount of restoration product reduced by 2.6-fold at the lowest concentration and 1.5-fold at the highest (*Figure 8—figure supplement 2B*).

Altogether these data show that both Fml1$^{\Delta C}$ and Fml1-MHF are particularly adept at catalysing the restoration of a lagging gap fork, which is a proposed intermediate of IFSA. The data also indicate that its ability to catalyse this reaction would not be impeded by the binding of RPA but would be inhibited by Rad52.

## Rad52 inhibits unwinding of a model IFSA junction by Fml1$^{\Delta C}$ and Fml1-MHF

To investigate whether Fml1 is capable of unwinding the putative IFSA junction (*Figure 7*, step 5a), we made a 5′ DNA flap substrate that mimics the annealing of a 3′ ssDNA tail into a ssDNA gap and tested whether Fml1$^{\Delta C}$ and Fml1-MHF can unwind it (*Figure 9A,B*). Both versions of Fml1 could unwind the 5′ flap and adjacent DNA strand, indicative of a 3′ to 5′ DNA helicase activity (*Figure 9B*). We next set up a reaction in which Rad52 was responsible for annealing the 5′ flap DNA strand into the ssDNA gap (*Figure 9C,D*). Having established this reaction, we tested whether Fml1 could unwind the annealed strand in the presence of Rad52 (*Figure 9C,E*). Both Fml1$^{\Delta C}$ and Fml1-MHF were strongly inhibited from unwinding the annealed strand when Rad52 was present, whereas they were still able to unwind the 'adjacent' DNA strand on the gapped substrate with no annealed 5′ flap (*Figure 9E*). The extent of inhibition by Rad52 in these reactions appeared to be greater than observed previously with the lagging gap fork (*Figure 8—figure supplement 2*). However, the conditions used for the Rad52 annealing reactions were quite different from those used in the fork restoration assays in terms of temperature, incubation time and concentrations of MgCl$_2$ and ATP. Therefore, we reassessed the effect of Rad52 on Fml1-MHF's ability to catalyse restoration of the lagging gap fork under conditions that more closely matched those used for the Rad52

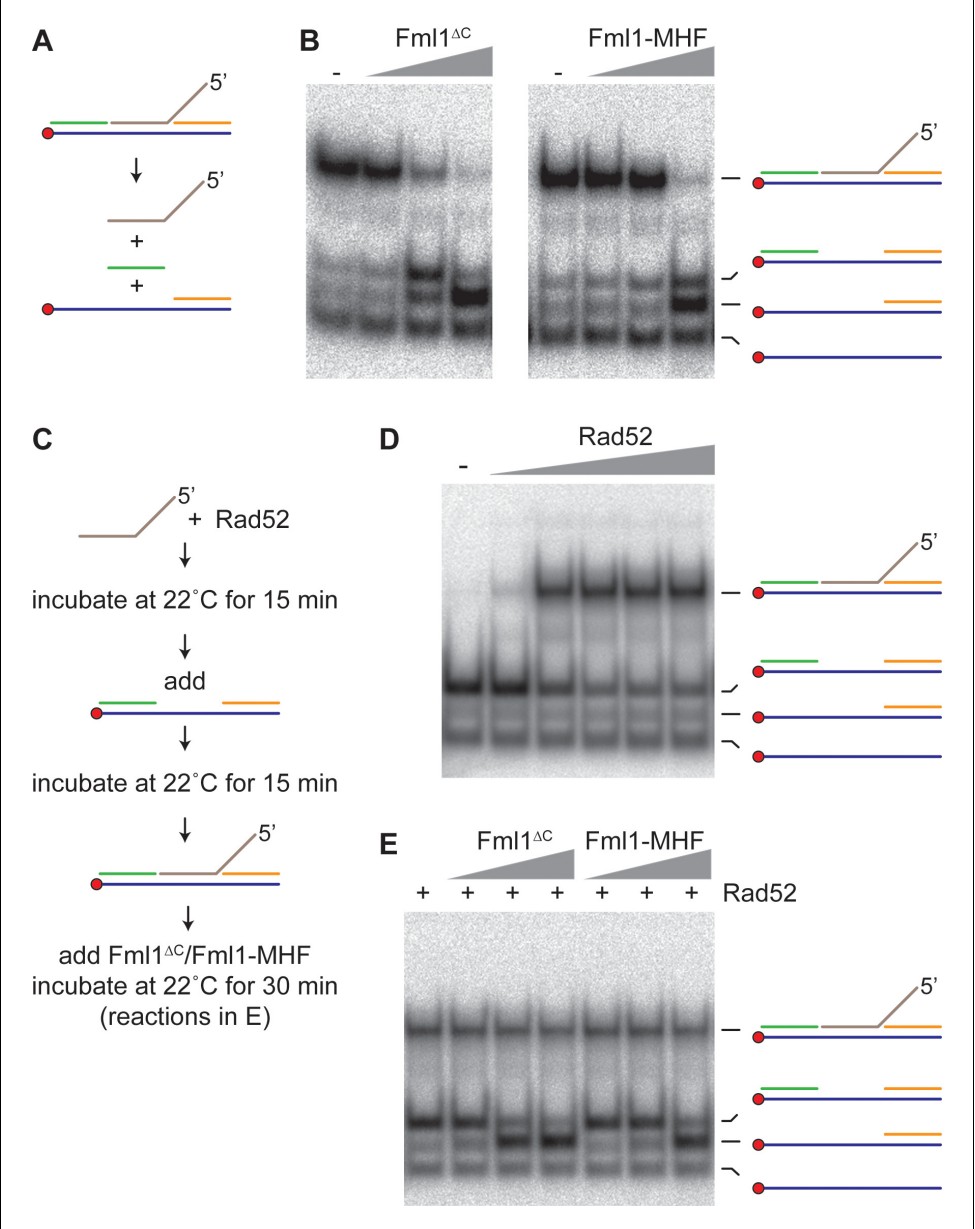

**Figure 9.** Effect of Rad52 on 5' flap unwinding by Fml1$^{\Delta C}$ and Fml1-MHF. (**A**) Schematic of the 5' flap DNA substrate and Fml1$^{\Delta C}$/Fml1-MHF reaction products. Component oligonucleotides are colour coded and the red circle indicates the 5' $^{32}$P label. (**B**) 5' flap (0.1 nM) unwinding by increasing concentrations (0.7 nM, 7 nM and 70 nM) of Fml1$^{\Delta C}$/Fml1-MHF. (**C**) Reaction scheme for Rad52 annealing reactions with and without Fml1$^{\Delta C}$/Fml1-MHF. (**D**) Annealing of 5' flap DNA strand (0.1 nM) into a ssDNA gap (0.1 nM) by increasing concentrations (20 nM, 50 nM, 100 nM, 200 nM and 310 nM) of Rad52. (**E**) Rad52 (150 nM) catalysed annealing of 5' flap DNA strand (0.1 nM) into a ssDNA gap (0.1 nM) plus addition of Fml1$^{\Delta C}$/Fml1-MHF (0.7 nM, 7 nM and 70 nM) following the reaction scheme in C.

The online version of this article includes the following figure supplement(s) for figure 9:

**Figure supplement 1.** Effect of Rad52 on lagging gap fork restoration by Fml1-MHF under similar reaction conditions as used in the experiment in *Figure 9E*.

annealing reactions (*Figure 9—figure supplement 1*). Under these conditions, Rad52 inhibition of fork restoration was noticeably less than seen in *Figure 8—figure supplement 2*. These data suggest that the difference in the extent of Rad52 inhibition of fork restoration and 5' flap unwinding by Fml1-MHF is not a consequence of the different reaction conditions used.

## Discussion

Our laboratory has previously reported that Fml1 promotes Rad51-dependent gene conversions induced by fork collapse at the *RTS1* barrier (*Sun et al., 2008*). We think it does this by catalysing fork reversal (*Figure 7*, step 1) and/or by unwinding D-loops formed by Rad51-mediated strand invasion, which would also limit deletions (*Figure 7*, steps 5* - 7*) (*Lorenz et al., 2012*; *Sun et al., 2008*). However, in addition to influencing Rad51-dependent recombination, Fml1 also suppresses genomic deletions that arise from Rad51-independent IFSA. Most telling is our finding that Rad51-independent SDDs increase by ~8 fold in a *fml1Δ* mutant (*Figure 1C*), which clearly establishes that Fml1's role in suppressing SDDs is distinct from its role in promoting Rad51-dependent gene conversions. We have shown that suppression of SDDs depends on Fml1's ATPase activity, suggesting that its DNA helicase/translocase activity is required. Fml1's non-catalytic C-terminal domain also supports SDD suppression. However, our observation that Fml1$^{\Delta C}$ retains some ability to suppress SDDs suggests that this domain is only needed to enhance or support the catalytic activity. This contrasts with Fml1's roles in promoting gene conversion and DNA repair, which are both totally dependent on the protein's C-terminal domain. We have re-affirmed that the C-terminal domain of Fml1, like in its human and budding yeast orthologues, mediates the interaction with MHF (*Bhattacharjee et al., 2013*; *Singh et al., 2010*; *Tao et al., 2012*; *Yan et al., 2010*). We have also shown that specific amino acid mutations within Fml1's C-terminal domain and Mhf2, which weaken Fml1-MHF complex formation in vitro and in vivo, phenocopy a *fml1$^{\Delta C}$* mutant with respect to SDD frequency, and partially phenocopy it with respect to gene conversion frequency and MMS hypersensitivity. These data show that the interaction between Fml1's C-terminal domain and MHF is important for Fml1's role in DNA repair and recombination.

Our finding that the level of Fml1-13Myc is dramatically reduced in *mhf2$^{D87A}$* cells suggests that a key function for Fml1's interaction with MHF is to promote its stability. Presumably MHF protects Fml1 from being targeted for removal by the proteasome by preventing its C-terminal domain from misfolding or by shielding a hypothetical degron. Indeed, it is interesting to speculate as to whether modulation of Fml1's interaction with MHF is used as a mechanism to regulate its level. It will also be interesting to ascertain how the AAA mutation in Fml1 enhances its stability.

Although clearly important, promoting protein stability is not the sole function for Fml1's interaction with MHF. This is evident from two observations: 1) Fml1$^{\Delta C}$ is defective in DNA repair and recombination whilst remaining relatively stable; and 2) the synergistic increase in MMS sensitivity of an *fml1$^{AAA}$ mhf2$^{D87A}$* double mutant correlates with reduced interaction between Fml1 and MHF in vitro but not further reduction in Fml1 stability in vivo. This second observation implies that there must be residual interaction between Fml1 and Mhf2$^{D87A}$ in vivo, which is sufficient to promote Fml1 activity without imparting protein stability. We also think that the increased stability of Fml1$^{AAA}$ may explain why it appears to be proficient in DNA repair despite exhibiting a weakened interaction with MHF in vitro.

So, what other role(s) for Fml1's interaction with MHF? In addition to promoting its stability, MHF may directly aid Fml1 by promoting its DNA binding and/or recruitment to collapsed replication forks. Indeed, it is known that the interaction between Fml1/FANCM and MHF synergistically enhances DNA binding through the creation of an additional DNA binding site (*Bhattacharjee et al., 2013*; *Singh et al., 2010*; *Tao et al., 2012*; *Yan et al., 2010*). It is thought that this enhanced DNA binding explains how FANCM-MHF exhibits a stronger fork regression activity than FANCM alone (*Singh et al., 2010*; *Yan et al., 2010*). We observed a similar difference between Fml1-MHF and Fml1$^{\Delta C}$ in driving the restoration of both lagging gap and regressed forks, with Fml1-MHF catalysing an ~3–4-fold faster reaction rate than Fml1$^{\Delta C}$. Although we cannot attribute this enhancement to MHF (due to an inability to purify and, therefore, test full-length Fml1 in the absence of MHF), it seems likely that the kinetics of fork restoration would be improved by MHF, either through the additional DNA binding that it brings to the complex, which could promote on-rate and processivity, and/or its potential ability to anneal complementary ssDNA (*Yan et al., 2010*), which could help

drive fork restoration. Enhanced DNA binding, especially to linear duplex DNA, may also explain how Fml1-MHF performs worse than Fml1$^{\Delta C}$ when restoring the lagging gap fork at sub-stoichiometric protein concentrations. In this case, turnover of the complex between substrates may be delayed by Fml1-MHF binding to the linear duplex reaction products.

Our finding that Fml1 can catalyse fork restoration in vitro provides a plausible explanation for how it might suppress SDDs in vivo. Essentially this reaction would deprive Rad52 of the ssDNA tail it would need to anneal into the lagging strand gap of the oncoming replication fork (*Figure 7*, steps 2a and 3a) (*Morrow et al., 2017*). It might even drive the displacement of an already annealed strand, prior to DNA junction cleavage by Mus81, which would abort the IFSA reaction and thereby prevent deletions (*Figure 7*, step 5b) (*Morrow et al., 2017*). Importantly Fml1 exhibits a strong preference for restoring a lagging gap regressed fork, which is precisely the type of substrate that is thought to form following fork collapse at the *RTS1* barrier (*Jalan et al., 2019*; *Morrow et al., 2017*; *Nguyen et al., 2015*; *Teixeira-Silva et al., 2017*). Moreover, its ability to catalyse this reaction is not impeded by RPA, meaning that it is capable of functioning under conditions similar to those expected in vivo (*Figure 7*, step 3a). However, our finding that Rad52 inhibits the restoration reaction suggests that Fml1's ability to act in vivo will be impeded by the binding of Rad52 to the regressed fork. Human RAD52 was recently shown to impede SMARCAL1 binding to stalled replication forks and thereby inhibit fork reversal (*Malacaria et al., 2019*). We suspect that Rad52 similarly inhibits fork restoration in fission yeast by occluding Fml1 from the regressed fork. The extent of fork restoration in vivo may therefore depend on the relative concentrations of Rad52 and Fml1 at the regressed fork, with optimal Fml1 activity occurring prior to Rad52 loading (i.e. steps 2a and 3a rather than 4a and 5b in *Figure 7*). Our in vitro data also indicate that Fml1 is unlikely to be very effective at directly unwinding the IFSA junction in vivo (*Figure 7*, step 5a), as this reaction seems to be even more strongly inhibited by Rad52 than fork restoration.

In comparison to a lagging gap fork, Fml1 exhibits relatively little activity on a leading gap regressed fork and, unlike the lagging gap fork, its ability to restore this substrate is inhibited by RPA. This would make sense in vivo, where a leading gap regressed fork is thought to be an intermediate of a particular type of DNA damage tolerance pathway, in which uncoupling of leading and lagging strand polymerases is provoked by a lesion in the leading template strand (e.g. a 3-methyl adenine induced by MMS), resulting in progression of the lagging strand polymerase beyond where the leading strand polymerase is blocked (*Higgins et al., 1976*; *Marians, 2018*; *Prado, 2018*). Fork regression in this scenario generates a 5' ssDNA tail, which can be used by the leading strand polymerase in a template switching reaction to bypass the blocking lesion. Various motor proteins, including Fml1 and its orthologues, have been implicated in catalysing this fork regression in vivo to promote template switching and consequent resistance to MMS (*García-Luis and Machín, 2018*; *Quinet et al., 2017*; *Sun et al., 2008*; *Whitby, 2010*; *Xue et al., 2015b*). Following lesion bypass, the regressed replication fork, which is now a fully duplex structure, has to be reset so that DNA replication can continue. Resetting may involve resection by a 5' to 3' exonuclease, such as Exo1, to generate a regressed fork with a 3' ssDNA tail that is amenable to restoration by a motor protein. One motor protein that has been implicated in driving this reaction is human SMARCAL1 (*Bétous et al., 2013*). However, our finding that Fml1-MHF is also adept at catalysing the restoration of a lagging gap regressed fork suggests that it too may contribute to fork resetting following template switching. A combination of fork regression followed by fork restoration may also account for FANCM's role in promoting the resumption of DNA synthesis after fork stalling induced by camptothecin in human cells (*Blackford et al., 2012*; *Luke-Glaser et al., 2010*; *Schwab et al., 2010*).

Similar to bacterial RecG and SMARCAL1 (*Bétous et al., 2013*), Fml1 is capable of catalysing both fork regression and restoration (*Nandi and Whitby, 2012*; *Sun et al., 2008*). The balance of these two opposing activities is presumably crucial in determining the extent to which Fml1 promotes and suppresses IFSA. Exactly how this balance is struck remains unclear, but it is likely that additional factors, including Rad52, will be instrumental in vivo. For example, in budding yeast, the fork regression activity of Fml1's orthologue Mph1 is inhibited by Smc5-Smc6 and promoted by Mte1 (*Silva et al., 2016*; *Xue et al., 2014*; *Xue et al., 2016*). In future studies, it will be important to investigate whether the orthologues of these proteins influence IFSA in fission yeast.

# Materials and methods

**Key resources table**

| Reagent type (species) or resource | Designation | Source or reference | Identifiers | Additional information |
|---|---|---|---|---|
| Strain, strain background (*S. pombe*) | MCW8020 | PMID: 28586299 | | standard laboratory strain (972) derivatives; see *Supplementary file 2* |
| Strain, strain background (*S. pombe*) | MCW8300 | PMID: 28586299 | | standard laboratory strain (972) derivatives; see *Supplementary file 2* |
| Strain, strain background (*S. pombe*) | MCW2080 | PMID: 18851838 | | standard laboratory strain (972) derivatives; see *Supplementary file 2* |
| Strain, strain background (*S. pombe*) | MCW5846, MCW5963 and MCW4406 | PMID: 24026537 | | standard laboratory strain (972) derivatives; see *Supplementary file 2* |
| Strain, strain background (*S. pombe*) | various strains | this paper | | standard laboratory strain (972) derivatives; see *Supplementary file 2* |
| Antibody | anti-GFP (Mouse monoclonal) | Clontech | RRID: AB_2313808 | dilution used (1:1000) |
| Antibody | anti-PCNA (Mouse monoclonal) | Santa Cruz Biotechnology | RRID: AB_628110 | dilution used (1:1000) |
| Antibody | anti-Myc tag (Goat polyclonal) | Abcam | RRID: AB_307033 | dilution used (1:10000) |
| Antibody | anti-Fibrillarin (38F3) (Mouse monoclonal) | Novus Biologicals | RRID: AB_2100980 | dilution used (1:1000) |
| Antibody | Myc-Trap Magnetic Agarose | Chromotek | RRID: AB_2631370 | amount used (20 µl) |
| Recombinant DNA reagent | various plasmids | this paper | | plasmid; see Materials and methods |
| Recombinant DNA reagent | pSN3 | PMID: 18851838 | | plasmid; see Materials and methods |
| Recombinant DNA reagent | pMW891 | PMID: 24026537 | | plasmid; see Materials and methods |
| Recombinant DNA reagent | pET19b-SpSSB | PMID: 8702843 | | plasmid; see Materials and methods |
| Recombinant DNA reagent | pMW601 | PMID: 15486206 | | plasmid; see Materials and methods |
| Sequence-based reagent | various oligonucleotides | this paper | | oligonucleotide; see Materials and methods |

*Continued on next page*

*Continued*

| Reagent type (species) or resource | Designation | Source or reference | Identifiers | Additional information |
|---|---|---|---|---|
| Sequence-based reagent | regressed fork substrate oligonucleotides | PMID: 23746452 | | oligonucleotide; see Materials and methods |
| Sequence-based reagent | X-12 substrate oligonucleotides | PMID: 9857040 | | oligonucleotide; see Materials and methods |
| Peptide, recombinant protein | Fml1$\Delta$C$^{1-603}$ | this study | | recombinant protein; see Materials and methods |
| Peptide, recombinant protein | Fml1-MHF | this study | | recombinant protein; see Materials and methods |
| Peptide, recombinant protein | Mhf1-Mhf2 | this study | | recombinant protein; see Materials and methods |
| Peptide, recombinant protein | Mhf1-Mhf2$^{D87A}$ | this study | | recombinant protein; see Materials and methods |
| Peptide, recombinant protein | RPA | this study | | recombinant protein; see Materials and methods |
| Peptide, recombinant protein | Rad52 | this study | | recombinant protein; see Materials and methods |

## *S. pombe* strains

*S. pombe* strains are listed in **Supplementary file 2**. The recombination reporter strain MCW8020 has been described previously (**Morrow et al., 2017**), and derivatives of this strain were constructed by standard genetic crosses. The *fml1*$^{\Delta C}$, *fml1*$^{\Delta C}$-*13Myc*, *fml1*$^{AAA}$ and *fml1*$^{AAA}$-*13Myc* strains were re-made using the method described previously (**Bhattacharjee et al., 2013**). The *mhf2$\Delta$::mhf2$^+$-kanMX6* strain was constructed by gene targeting using a linear DNA cassette excised from pMW915 using SalI and EcoRV. The *mhf2* point mutant strains were constructed by gene targeting using derivatives of pMW915 (pJN2, pJN3 and pJN4) containing appropriate mutations introduced by site-directed mutagenesis. Strains were verified by phenotypic tests, diagnostic PCRs and DNA sequencing as appropriate.

## Plasmids

Plasmid pMW915 is a derivative of pFA6a-kanMX6 (**Bähler et al., 1998**) containing the *mhf2* 5' UTR and open reading frame, and *mhf2* 3' UTR amplified from *S. pombe* genomic DNA using oligonucleotides oMW1431 (5'-ATAGTCGACTCGTTCAATGCTGCCGGCTG-3') and oMW1444 (5'-ATGGCGCGCCCTAACTAAAGTCAAGGGCTAG-3'), and oMW1433 (5'-TAGAGCTCGCTGATACTAAATGGAGACG-3') and oMW1434 (5'-TTGGATATCACCCCAAAGCACTTATC-3'), respectively. pJN2, used for the construction of the *mhf2$^{S59A}$::kanMX6* strain, was made by site-directed mutagenesis of pMW915 using oMW1507 (5'-TATGAAGAGAAAAAGAACGCAATCATGTCATCTTCTGAA-3') and oMW1508 (5'-TTCAGAAGATGACATGATTGCGTTCTTTTTCTCTTCATA-3'). pJN3, used for the construction of the *mhf2$^{Q83A}$::kanMX6* strain, was made by site-directed mutagenesis of pMW915 using oMW1517 (5'-GAAAATGGCATCGCAGCTGCACTAGCCCTTGACTTTAGT-3') and oMW1518 (5'-ACTAAAGTCAAGGGCTAGTGCAGCTGCGATGCCATTTTC-3'). pJN4, used for the construction of the *mhf2$^{D87A}$::kanMX6* strain, was made by site-directed mutagenesis of pMW915 using oMW1519 (5'-GCAGCTCAACTAGCCCTTGCATTTAGTTAGGGCGCGCCA-3') and oMW1520 (5'-TGGCGCGCCCTAACTAAATGCAAGGGCTAGTTGAGCTGC-3'). The plasmids for expressing fragments of Fml1 fused to MBP are derivatives of pMAL-c5X (New England BioLabs) with the appropriate *fml1* gene fragment inserted between the BamHI and SbfI sites in the plasmid. These plasmids are: pIN28 (Fml1$^{1-603}$); pIN31 (Fml1$^{1-575}$); pIN27 (Fml1$^{576-603}$); pIN25 (Fml1$^{576-690}$); pIN26

(Fml1$^{576-725}$); pIN24 (Fml1$^{576-834}$); pIN38 (Fml1$^{401-730}$); and pIN33 (Fml1$^{604-834}$). pIN37 and pIN34, which express Y672A-R674A-R678A mutant Fml1 fragments, were derived from pIN24 and pIN33, respectively, by site-directed mutagenesis. The plasmids for expressing His-tagged Fml1ΔC$^{1-603}$ (pSN3), Mhf1 with His-tagged Mhf2 (pMW891), RPA (pET19b-SpSSB) and Rad52 (pMW601) have been described (*Bhattacharjee et al., 2013*; *Doe et al., 2004*; *Ishiai et al., 1996*; *Sun et al., 2008*). The plasmid for co-expression of Mhf1 and His-tagged Mhf2$^{D87A}$ was made by site-directed muta-genesis of pMW891. The plasmid for co-expression of full-length Fml1 together with Mhf1 and His-tagged Mhf2 (pJBB77) was constructed by inserting the T7 phage Ø10 promoter plus full-length *fml1* gene, amplified from a pT7-7 derivative, at the NheI site in pMW891. All plasmids were verified by DNA sequencing.

## Media and genetic methods

Protocols for the growth and genetic manipulation of *S. pombe*, spot assays and assays for recombination have been described (*Jalan et al., 2019*; *Morrow et al., 2017*; *Nguyen et al., 2015*; *Osman and Whitby, 2009*; *Tamang et al., 2019*). Recombination experiments were repeated at least twice with between 5 and 10 colonies being assayed in each experiment. Strains being directly compared were analysed at the same time in parallel experiments. For spot assays, a 10-fold dilution series was plated for each strain ranging from 10$^5$ to 10$^2$ cells except for *Figure 3B* where the series ranged from 10$^6$ to 10$^2$ cells. Spot assay plates were incubated at 30 ˚C for 3 to 4 days before being photographed. Each spot assay was performed at least twice, using independent cell cultures, to confirm that the data were reproducible. Statistical analysis of recombination data was performed in GraphPad Prism Version 8.0.2. Due to some of the recombination data not conforming to a normal distribution, comparisons were made using the Mann-Whitney U test. Sample sizes and *p* values are given in *Supplementary file 1*.

## Microscopy

For analysis of chromosome segregation, cells from an exponentially growing culture in Yeast Extract plus supplements (YES) were harvested and fixed with 70% ethanol. The fixed cells were then stained with Calcofluor White and SYBR Green I prior to imaging with an Olympus BX50 epifluorescence microscope equipped with filters to detect blue and green fluorescence (Chroma Technology, VT). For analysis of Mhf1-GFP, cells from an exponentially growing culture in YES were washed in water and then immediately stained with 4′,6-diamidino-2-phenylindole (DAPI) and mounted in Vectashield (Vector Laboratories). The cells were then imaged using an Eclipse TE2000-U microscope (Nikon), equipped with a X100/1.4 oil PlanApo objective lens and appropriate filter sets to detect blue, green and red fluorescence. Black and white images were acquired with a CoolSNAP HQ$^2$ camera (Photometrics) controlled by MetaMorph v.7.7.3.0 software (Molecular Devices, CA). Images were pseudo-coloured and overlayed using Adobe Photoshop CS5 (Adobe Systems, CA).

## Antibodies, western blotting and Immunoprecipitation

Living Colors GFP mouse monoclonal antibody (1:1000) (Clontech Laboratories, CA), PCNA (PC10) mouse monoclonal antibody (1:1000) (Santa Cruz Biotechnology), anti-Fibrillarin mouse monoclonal antibody (1:1000) (Novus Biologicals) and anti-Myc tag goat polyclonal antibody (1:10000) (Abcam) were used in western blotting. For analysis of Mhf1-GFP and Fml1-13Myc levels, whole cell extracts were prepared from cultures growing exponentially in YES at 30 ˚C. Cells were harvested by centrifugation and washed in ice cold stop solution (150 mM NaCl, 50 mM NaF, 10 mM EDTA, 1 mM NaN$_3$). After further centrifugation, approximately 100 μl of pelleted cells were resuspended in 200 μl of water. An equal volume of 600 mM NaOH was then added to the mixture, which was then incubated at room temperature for 10 min. The cells were then pelleted by centrifugation and lysed by resuspending in 100 μl of SDS loading dye and boiling for 3 min. Following further centrifugation at 21,000 x g for 2 min, the supernatant was saved for analysis by western blotting.

For immunoprecipitation of Fml1-13Myc, cell cultures growing exponentially in YES at 30 ˚C were treated with 1 mM NaN$_3$ on ice to arrest growth. The cells were then harvested by centrifugation, washed once with water and then with IP buffer (50 mM Tris-HCl [pH 7.5], 150 mM NaCl). Cells were then resuspended in an equal volume of IP lysis buffer (50 mM Tris-HCl [pH 7.5], 150 mM NaCl, 1 mM EDTA, 1% Triton X-100, 1 mM PMSF, 1 x complete EDTA-free protease inhibitor cocktail

[Roche], 1 x Pierce Phosphatase Inhibitor [Thermo Scientific]) and lysed by four cycles (30 s each) of bead beating at 4 ˚C using an equal volume of glass beads (~5 mm diameter) and a vortex mixer. Following removal from the glass beads, the lysate was centrifuged at 21,000 x g for 15 min at 4 ˚C and the supernatant was then extracted and stored on ice. The lysate pellet was then resuspended in 0.1 ml of Benzonase buffer (50 mM Tris-HCl [pH 8.0], 20 mM NaCl, 2 mM MgCl$_2$) plus 1 µl of Benzonase ($\geq$250 units/µl) (Sigma) and incubated at room temperature for 45 min. Following further centrifugation at 21,000 x g for 15 min at 4 ˚C, the soluble fraction from the Benzonase treated sample was added to the rest of the soluble lysate and incubated with 20 µl Myc-Trap magnetic agarose beads (Chromotek) on a tube rotator at 4 ˚C for 90 min. The agarose beads were then magnetically separated from the supernatant, washed with IP buffer (5 × 1 ml) and resuspended in 2 x SDS-sample buffer (120 mM Tris-HCl [pH 6.8], 20% glycerol, 4% SDS, 0.04% bromophenol blue; 10% β-mercaptoethanol) before boiling at 95 ˚C for 10 min to dissociate immunoprecipitated protein.

## Proteins

His-tagged Fml1ΔC$^{1-603}$ and the Mhf1 plus His-tagged Mhf2 complex were expressed from plasmids pSN3 and pMW891, respectively, in *E. coli* BL21-CodonPlus (DE3)-RIL cells (Agilent) and purified as described (*Bhattacharjee et al., 2013*; *Sun et al., 2008*). The Mhf1 plus His-tagged Mhf2$^{D87A}$ complex was purified in exactly the same way as the wild-type complex. *S. pombe* RPA was expressed from plasmid pET19b-SpSSB in *E. coli* BL21-CodonPlus (DE3)-RIPL cells (Agilent) and purified as described (*Haruta et al., 2006*). The purification of the Fml1-MHF complex, MBP-Fml1 fusions and His-tagged Rad52 are described below. Protein concentrations were estimated using a Bio-Rad protein assay kit with bovine serum albumin as the standard. Amounts of His-tagged Fml1ΔC$^{1-603}$, MBP-Fml1 fusions and His-tagged Rad52 are expressed in moles of monomer, whereas Fml1-MHF, Mhf-Mhf2 and RPA are expressed in moles of complex based on the following stoichiometries: Fml1(1)-Mhf1(2)-Mhf2(2); Mhf1(2)-Mhf2(2); Ssb1(1)-Ssb2(1)-Ssb3(1).

## Expression and purification of MBP-Fml1 fragments

The various fragments of Fml1 fused to MBP were expressed from the appropriate plasmid (pIN28, pIN31, pIN27, pIN25, pIN26, pIN24, pIN37, pIN38, pIN33 and pIN34) in *E. coli* BL21-CodonPlus (DE3)-RIL cells (Agilent). Cultures were grown in Luria-Bertani (LB) broth supplemented with 50 µg/ml of ampicillin and 20 µg/ml of chloramphenicol at 25˚C. At an OD$_{600}$ of 0.6, isopropyl-β-D-thiogalactopyranoside (IPTG) was added to a final concentration of 0.5 mM to induce MBP-tagged protein expression at 25˚C for 5 hr. The cells were then harvested by centrifugation and stored at −80 ˚C until required. All of the subsequent steps were at 4 ˚C. Cells (~10 g) were thawed, resuspended in 3 ml/g of lysis buffer (25 mM Tris-HCl [pH 8.0], 1 mM EDTA, 2.5 mM dithiothreitol [DTT], 1% v/v Triton X-100, 1 mM phenylmethylsulfonyl fluoride [PMSF], 10% v/v glycerol) plus 1M NaCl, disrupted by sonication and then centrifugated at 40,000 x g for 30 min. The cleared supernatant was incubated with amylose agarose beads (1 ml per 10 g of cells, New England Biolabs) for 2 hr. Beads were loaded into a column and washed sequentially with 40 bead volumes of lysis buffer plus 1M NaCl and 40 bead volumes of lysis buffer plus 0.2 M NaCl. The protein fragments were eluted with elution buffer (25 mM Tris-HCl [pH 8.0], 0.3M NaCl, 1 mM EDTA, 2.5 mM DTT, 1% v/v Triton X-100, 1 mM PMSF, 20 mM Maltose, 10% v/v glycerol). The protein eluates were further purified by gel filtration chromatography (HiLoad 16/60 Superdex 200 pg, GE Healthcare) in buffer (20 mM Tris-HCl [pH 8.0], 2 mM DTT, 0.1 mM EDTA, 0.1 mM PMSF, 0.1% v/v Triton X-100, 10% v/v glycerol) plus 300 mM NaCl. The peak fractions were pooled, aliquoted and stored at −80 ˚C.

## Expression and purification of Fml1-MHF

Full-length Fml1, Mhf1 and His-tagged Mhf2 (Fml1-MHF) were co-expressed from pJBB77 in *E. coli* BL21-CodonPlus (DE3)-RIL cells (Agilent). Cultures were grown in LB broth supplemented with 50 µg/ml of carbenicillin and 20 µg/ml of chloramphenicol at 25 ˚C. At an OD$_{600}$ of 0.6, IPTG was added to a final concentration of 0.5 mM to induce Fml1-MHF expression at 25 ˚C for 5 hr. Cells were harvested by centrifugation and stored at −80 ˚C until required. All of the subsequent steps were at 4 ˚C. Cells (~10 g) were thawed, resuspended in 20 ml of Buffer H (50 mM potassium phosphate [pH 8.0], 300 mM NaCl, 10% v/v glycerol) plus 5 mM β-mercaptoethanol, 0.5% v/v Triton X-100 and protease inhibitor cocktail (one tablet Roche EDTA-free protease inhibitor/50 ml), and disrupted by

passage through a French pressure cell at 19,000 p.s.i.. Cell debris was then removed by centrifugation at 40,000 x g for 30 min and the supernatant was loaded onto a 1 ml gravity flow nickel-nitrilotriacetic acid (Ni-NTA) Superflow column (Qiagen). The column was washed with 60 column volumes of Buffer H plus 20 mM imidazole and then bound protein was eluted with three column volumes of Buffer H plus 200 mM imidazole. Eluted protein was loaded onto a HiLoad 16/60 Superdex 200 pg, gel filtration column (GE Healthcare) which was then developed with 120 ml of Buffer A (50 mM Tris-HCl [pH 8.0], 1 mM EDTA, 1 mM DTT, 1 mM PMSF, 10% v/v glycerol) plus 300 mM NaCl. To generate Fml1-MHF samples for crosslinking mass spectrometry analysis, the Tris-HCl in Buffer A was substituted with 25 mM HEPES-NaOH [pH 7.5]. Peak Fml1-MHF fractions were pooled, aliquoted and stored at −80 °C.

## Expression and purification of Rad52

His-tagged *S. pombe* Rad52 was expressed from plasmid pMW601 in *E. coli* BL21-CodonPlus (DE3)-RIL cells (Agilent) as described (*Doe et al., 2004*). Cells lysis and Rad52 purification by Ni-NTA and gel filtration chromatography were the same as described for Fml1-MHF above. Gel filtration column fractions containing the peak of Rad52 protein were pooled, diluted three-fold in Buffer A and applied to a 1 ml HiTrap Q HP column (GE Healthcare). Bound protein was then eluted with a linear gradient of 0.1 to 1.0 M NaCl in Buffer A. Fractions containing the peak of Rad52 protein were pooled, diluted three-fold in Buffer A and applied to a 1 ml HiTrap Heparin HP column (GE Healthcare). Bound protein was eluted with a linear gradient of 0.1 to 1.0 M NaCl in Buffer A and the peak Rad52 fractions were pooled, concentrated and exchanged to Buffer A plus 0.3 M NaCl before aliquoting and storing at −80 °C.

## Pull-down assay

Approximately 32 pmol of purified MBP-tagged Fml1 fragment (or 64 pmol of MBP-Fml1[576-834]$^{AAA}$ or 96 pmol MBP-Fml1[604-834]$^{AAA}$) was incubated with 10 µl of amylose beads (New England Biolabs) in Buffer I (25 mM Tris-HCl [7.5], 0.1 mM EDTA, 0.01% v/v NP40, 0.1 mM PMSF, 2 mM DTT, 10% v/v glycerol) by gentle rotation for 2 hr at 4°C. After removing the unbound protein by washing with Buffer I (with NaCl as indicated) three times, purified Mhf1-Mhf2 (wild-type or D87A mutant) was added to the beads with bound protein and incubated for 3 hr at 4°C. After three consecutive 20 min washes with Buffer I plus NaCl as indicated, bound proteins were eluted with 28 µl of 2% SDS and analysed by SDS-PAGE. Technical replicates were performed for each pull-down assay to ensure the results were reproducible.

## Crosslinking mass spectrometry

Protein crosslinking was performed as described (*Leitner et al., 2014*). After establishing the optimal concentration of the crosslinker to avoid over-crosslinking, 45 µg of purified Fml1-MHF was crosslinked with 0.16 mM of H12- and D12-labelled BS3 (Creative Molecules) at 37 °C for 30 min. The crosslinking reaction was then quenched by adding 1M NaHCO$_3$ (50 mM final concentration) and incubating for a further 20 min at 37 °C. It was then evaporated in a vacuum centrifuge, re-dissolved in 6 M urea and treated with TCEP for 30 min at 37 °C to reduce disulphide bonds. This was followed by treatment with iodoacetamide in the dark at room temperature for 30 min. The mixture was then diluted with 50 mM NH$_4$HCO$_3$ to achieve a final concentration of 1 M urea before digesting overnight with trypsin. The sample was then desalted, evaporated to dryness, resuspended in 70% H$_2$O/30% ACN/0.1% TFA and applied to a Superdex Peptide 10/300 GL gel filtration column (GE Healthcare). Peak fractions of crosslinked peptides were pooled and analysed by liquid chromatography tandem mass spectrometry (LC-MS/MS) using an Orbitrap Elite (Thermo Scientific, Waltham, MA) as described (*Adam et al., 2011*). XQuest software was used to analyse the MZXML data files (*Leitner et al., 2014*) and the results were displayed as a Circos-like plot using a confidence score cut-off >15 based on a FDR < 10%.

## EMSA, fork restoration, strand annealing and unwinding assays

The $^{32}$P-labelled DNA substrates were generated by annealing oligonucleotides followed by gel purification as described previously (*Bétous et al., 2013*; *Whitby and Dixon, 1998*). The 5' flap substrate ('strand annealing substrate') was constructed from oligonucleotides oMW305 (5'-GACGC

TGCCGAATTCTACCAGTGCCTTGCTAGGACATCTTTGCCCACCTGCAGGTTCACCC-3'), oMW308 (5'-TAAGAGCAAGATGTTCTATAAAAGATGTCCTAGCAAGGCAC-3'), oMW1657 (5'-TGGTAGAA TTCGGCAGCGTC-3') and oMW1658 (5'-GGGTGAACCTGCAGGTGGGC-3'). The binding reactions contained 0.5 nM $^{32}$P-labelled X-12 in buffer (25 mM Tris-HCl [pH 8.0], 1 mM DTT, 0.1 mg/ml BSA, 45 mM NaCl, 0.1% v/v NP40 and 6% v/v glycerol) plus the indicated amount of Mhf1-Mhf2 or Mhf1-Mhf2$^{D87A}$ protein. Reactions were incubated on ice for 15 min before being resolved on a 4% (19:1) polyacrylamide gel in 0.25X TBE buffer. Fork restoration reactions contained 0.5 nM of $^{32}$P-labelled fork substrate in buffer (40 mM Tris-HCl [pH 8.0], 2 mM DTT, 90 mM KCl, 15 mM NaCl, 0.1 mg/ml BSA, 5 mM MgCl$_2$ and 2 mM ATP), except for the reactions in *Figure 9—figure supplement 1*, which contain 'strand annealing' buffer (25 mM Tris-HCl [pH 8.0], 1 mM DTT, 0.1 mg/ml BSA, 6% glycerol, 45 mM NaCl, 2.5 mM MgCl$_2$ and 5 mM ATP). Reactions were started by the addition of Fml1$^{\Delta C}$/Fml1-MHF and incubated at 37°C for 20 min (or as indicated). For the reactions containing RPA, the reaction mixture was pre-incubated with RPA (20 nM) at 22°C for 15 min before addition of Fml1$^{\Delta C}$/Fml1-MHF. For fork restoration reactions containing Rad52, the reaction mixture was pre-incubated with Rad52 (210 nM) at 22°C for 10 or 15 min before addition of Fml1-MHF. The unwinding reactions in *Figure 9B* contained 0.1 nM of $^{32}$P-labelled 5'flap substrate in strand annealing buffer plus the indicated amount of Fml1$^{\Delta C}$/Fml1-MHF and were incubated at 22°C for 30 min. The strand annealing/unwinding reactions in *Figure 9D and E* contained 0.1 nM of unlabelled oligonucleotide oMW308 and 0.1 nM of $^{32}$P-labelled single-strand gap DNA (oligonucleotides oMW305, oMW1657 and oMW1658) in strand annealing buffer. All fork restoration and strand annealing/unwinding assays were stopped by addition of 5X termination buffer (2.5% SDS, 200 mM EDTA, 10 mg/ml proteinase K and 50% glycerol). Terminated reactions were resolved on 8% (19:1) polyacrylamide gels in 1X TBE buffer. Gels were dried on 3 MM Whatman paper, exposed to a Phosphor screen which was then scanned by a Fuji FLA3000 PhosphorImager. The digitised data was then analysed using ImageGauge V4.21 software (Fuji). Reactions were quantified by determining the substrate and product band intensities, which were normalized relative to the no protein control. The following formula was then applied: % reaction product = $(p^1 - p^0)/(p^1 - p^0 + s^1) \times 100$ where $p^1$ is the normalized product band intensity, $p^0$ is the product band intensity from the no protein control reaction and $s^1$ is the normalized substrate band intensity. All fork restoration, strand annealing and unwinding assays were repeated at least once to ensure that results were reproducible.

## Acknowledgements

We thank the late Jerard Hurwitz for the gift of pET19b-SpSSB, Carl Morrow for constructing the fission yeast strain MCW8296, Sonali Bhattacharjee for constructing plasmid pJBB77 and Nathan Rose for his help in performing the crosslinking mass spectrometry experiment.

## Additional information

### Funding

| Funder | Grant reference number | Author |
| --- | --- | --- |
| Biotechnology and Biological Sciences Research Council | BB/P019706/1 | Matthew C Whitby |
| Wellcome | 090767/Z/09/Z | Matthew C Whitby |

The funders had no role in study design, data collection and interpretation, or the decision to submit the work for publication.

### Author contributions

Io Nam Wong, Jacqueline PS Neo, Formal analysis, Investigation, Methodology, Writing - review and editing; Judith Oehler, Fekret Osman, Formal analysis, Supervision, Investigation, Methodology, Writing - review and editing; Sophie Schafhauser, Formal analysis, Investigation; Stephen B Carr, Investigation, Writing - review and editing; Matthew C Whitby, Conceptualization, Formal analysis, Supervision, Funding acquisition, Investigation, Methodology, Writing - original draft, Writing - review and editing

**Author ORCIDs**
Io Nam Wong https://orcid.org/0000-0002-4500-1758
Judith Oehler http://orcid.org/0000-0002-8397-6492
Matthew C Whitby https://orcid.org/0000-0003-0951-3374

**Decision letter and Author response**
Decision letter https://doi.org/10.7554/eLife.49784.sa1
Author response https://doi.org/10.7554/eLife.49784.sa2

## Additional files

### Supplementary files

- Supplementary file 1. Direct repeat recombinant frequencies.
- Supplementary file 2. *Schizosaccharomyces pombe* strains.
- Transparent reporting form

### Data availability

All data generated or analysed during this study are included in the manuscript and supporting files.

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
