## [Decision Letter]

**Acceptance summary:**

This work examines the role of FANCM-related helicase of *S. pombe*, Fml1, and its interaction with centromeric proteins Mhf1 and Mhf2 (CEMP-S, CEMP-X) in promoting genetic stability and genotoxin tolerance. This manuscript builds on the authors' previous work that described a novel Rad51-independent mechanism for genomic deletions during DNA replication, termed inter-fork strand annealing (IFSA). IFSA requires the Rad52 strand annealing protein and Mus81-Eme1 endonuclease and is suppressed by Fml1 helicase. The current study shows that suppression of IFSA requires Fml1 catalytic activity and is partially dependent on the C-terminal regulatory domain. The authors propose that the catalytic activity of Fml1 drives fork restoration to prevent deletions. In the revised manuscript, the authors also show that Rad52 inhibits fork restoration by Fml1-MHF in vitro. The authors identified a mhf2 mutant that is partially defective for Fml1 interaction and MMS resistance, and appears to retain full centromere function. When *mhf2^D87A^* is combined with the previously described *fml1^AAA^* mutation, which partially impairs interaction with Mhf1-Mhf2, the double mutant exhibits a similar phenotype to the mhf1 null for MMS sensitivity, and is similar to fml1 δ C for deletion formation. Furthermore, they show that Fml1 and Fml1 deltaC catalyze fork restoration in vitro, though activity is lower for the truncated protein.

The work has been substantially revised after the initial submission; the new data provide a strong support for the authors' model. All previous critiques were adequately addressed.

**Decision letter after peer review:**

Thank you for submitting your article "The Fml1-MHF complex suppresses inter-fork strand annealing in fission yeast" for consideration by *eLife*. Your article has been reviewed by three peer reviewers, and the evaluation has been overseen by a Reviewing Editor and Kevin Struhl as the Senior Editor. The reviewers have opted to remain anonymous.

The reviewers have discussed the reviews with one another and the Reviewing Editor has drafted this decision to help you prepare a revised submission.

The manuscript has been reviewed by three experts who were positive about the overall quality of work and concluded that it provides new interesting information. All three reviewers, however, raised some concern over whether the *mhf2^D87A^* is a clean separation-of-function mutation. The experiments suggested by the reviewers are quite doable within the revision time frame, and can clarify some of the main claims of the work:

Major points that need to be addressed by additional experiments:

1) (Figure 3—figure supplement 3B) Mhf1-GFP foci look very different in wild type and *mhf2^D86A^* cells, and the cells appear elongated. The authors suggest the additional nuclear staining of Mhf1-GFP in wild-type cells is due to its co-localization with Fml1. The authors should verify this conclusion by analyzing Mhf1-GFP in fml1 null or fml1 deltaC cells. Whether *mhf2^D87A^* is indeed a clean separation of function mutant needs to be addressed by checking endogenous protein level and co-IP in cells with Fml1 and *fml1^AAA^*. To be certain that *mhf1^D87A^* is a clean separation of function allele, it will be also useful to test whether it affects centromere functions (not just CEN localization).

2) Main prediction of the proposed model is that Fml1-MHF removes Rad52 from regressed forks. Can this be directly demonstrated in the same set up?

3) Subsection “Identification of a mutation in Mhf2 that disrupts the interaction between Fml1 and MHF” and Figure 3B: the epistasis claimed is not supported by these data, since the *mhf2^D87A^* mutant shows no sensitivity to MMS at these doses (and therefore epistasis can't be measured). This experiment should be done at 0.01% MMS, where it is sensitive.

Major points that need be addressed in revision:

1) (Figure 6) The authors compare biochemical activities of Fml1-MHF and Fml1 deltaC. The lagging gap and regressed forks are the most likely substrates in vivo and both are efficiently restored by Fml1-MHF. Fml1C exhibits higher activity than Fml1-MHF on the lagging gap substrate at sub-stoichiometric concentration, but at stoichiometric concentration, it restores the fork more slowly than Fml1-MHF. Given the high activity of Fml1delta C on the lagging strand substrate, it is surprising that it is not very effective in suppressing deletions. Is Fml1delta C recruited to the stalled fork?

2) Abstract: "enhanced by Mhf1 and Mhf2" this claim is too strong since the fork restoration activity of Fml1 in the absence of Mhf1/2 isn't done.

3) Subsection “Fml1 suppresses Rad51-independent SDDs”: it does look like there are some Rad51-dependent deletions as well, please comment

4) Subsection “Identification of a mutation in Mhf2 that disrupts the interaction between Fml1 and MHF” paragraph two: It is surprising by the extent of mobility change conferred by the D87A Mhf2 mutant. It is interpreted as being a subtle, separation-of-function mutant, but is seems likely there is a gross conformational change. This should be acknowledged.

5) Paragraph three in the same section: "speckled" this looks diffuse to me plus the signal from Mhf1 is more extensive than the DAPI. Explain?

6) Discussion in general: Is there some consequence/advantage of recruiting Fml1 to the centromere? Or do the authors think that Fml1 only plays a role at Mhf1/2 binding to noncentromeric sites?

---

## [Author Response]

Major points that need to be addressed by additional experiments:1) (Figure 3—figure supplement 3B) Mhf1-GFP foci look very different in wild type and mhf2^D86A^ cells, and the cells appear elongated. The authors suggest the additional nuclear staining of Mhf1-GFP in wild-type cells is due to its co-localization with Fml1. The authors should verify this conclusion by analyzing Mhf1-GFP in fml1 null or fml1 deltaC cells. Whether mhf2^D87A^ is indeed a clean separation of function mutant needs to be addressed by checking endogenous protein level and co-IP in cells with Fml1 and fml1^AAA^. To be certain that mhf1^D87A^ is a clean separation of function allele, it will be also useful to test whether it affects centromere functions (not just CEN localization).

We have revised Figure 3—figure supplement 3B to include data for Mhf1-GFP localization in a *fml1*∆ mutant. The *mhf2*^D87A^ mutant does not have an elongated cell phenotype and, therefore, we have used a different example image that is more representative of the overall cell population. The data show that in wild-type cells Mhf1-GFP forms foci that co-localize with Mis6-mCherry at the centromeres plus a more diffuse pattern of fluorescence throughout the rest of the nucleus. However, whilst Mhf1-GFP centromeric foci are observed, the diffuse pattern of nuclear fluorescence is absent or greatly diminished in *mhf2*^D87A^and *fml1*∆ mutant cells.

Unfortunately, we have been unable to directly measure the level of Mhf2^D87A^ in cells as epitope tagged constructs are not functional and antibodies raised against Mhf2 have so far been ineffective in western blots. Therefore, we have indirectly assessed the levels of Mhf2^D87A^ by determining the levels of its obligate partner Mhf1. In Figure 3—figure supplement 3A we show that the levels of Mhf1-GFP are greatly diminished in a *mhf2*∆ mutant whereas the same is not true in either a *mhf2*^D87A^ or *fml1*∆ mutant. The accompanying text describing these new data is in subsection “Identification of a mutation in Mhf2 that causes MMS hypersensitivity but does not disrupt MHF’s centromeric function” of the revised manuscript.

To determine the effect of *mhf2*^D87A^ on the interaction between Fml1 and MHF in vivo, we have performed a co-IP experiment using Fml1-13Myc/Fml1^AAA^-13Myc and Mhf1-GFP (Figure 4—figure supplement 1). Unlike *mhf2*^+^ cells, Mhf1-GFP is not detected in Fml1-13Myc or Fml1^AAA^-13Myc immunoprecipitates from *mhf2*^D87A^ mutant cells. The accompanying text describing these new data is in subsections “A D87A mutation in Mhf2 disrupts the interaction between Fml1 and MHF” and “Combining Fml1^AAA^ and Mhf2^D87A^ mutants greatly impairs Fml1^604-834^-MHF complex formation in vitro and causes a synergistic increase in MMS sensitivity in vivo” of the revised manuscript.

To determine whether the D87A mutation in Mhf1 disrupts its centromeric function, we compared chromosome segregation in wild-type, *mhf2*∆, *mhf2*^D87A^ and *fml1*∆ strains (Figure 3—figure supplement 1 and described in subsection “Identification of a mutation in Mhf2 that causes MMS hypersensitivity but does not disrupt MHF’s centromeric function”). These data show that chromosome segregation is severely disrupted in a *mhf2*∆ mutant but is relatively normal in a *mhf2*^D87A^ mutant.

In addition to the experiments requested by the reviewers (see above), we have also investigated the effect of *mhf2*^D87A^ on the stability of Fml1-13Myc and Fml1^AAA^-13Myc. In new data, presented in Figure 6, we show that the levels of both Fml1-13Myc and Fml1^AAA^-13Myc are greatly diminished in whole cell extracts from *mhf2*^D87A^ mutant cells compared to *mhf2^+^* cells. These data provide evidence that Fml1 stability depends on its interaction with MHF in vivo. The levels of Fml1^∆C^-13Myc and Fml1-13Myc are similar in *mhf2^+^* cells indicating that the instability of Fml1 is mediated by its C-terminal domain. Intriguingly, we also show that the levels of Fml1^AAA^-13Myc are much higher than Fml1-13Myc in both *mhf2^+^* and *mhf2*^D87A^ cells. Altogether these data provide important new insights into the function of Fml1’s C-terminal domain and interaction with MHF, which are described in subsection “A fml1^AAA^ mhf2^D87A^ double mutant exhibits reduced gene conversions and increased SDDs” and the Discussion section.

2) Main prediction of the proposed model is that Fml1-MHF removes Rad52 from regressed forks. Can this be directly demonstrated in the same set up?

We have not been able to directly determine whether Fml1-MHF removes Rad52 from regressed forks. However, we have investigated whether Rad52 inhibits fork restoration by Fml1-MHF (Figure 8—figure supplement 2 and Figure 9—figure supplement 1). We show that a large molar excess of Rad52 can partially inhibit restoration of a lagging gap fork by Fml1-MHF in vitro. We also present new experiments in which we have investigated whether Fml1^∆C^/Fml1-MHF can unwind a DNA strand annealed by Rad52 into a ssDNA gap, which is a mimic of the proposed IFSA junction (Figure 9). We find that DNA unwinding of the annealed strand by either Fml1^∆C^ or Fml1-MHF is strongly inhibited in the presence of Rad52. Moreover, the level of inhibition appears to be greater than seen in the fork restoration reactions. These new data are described in subsections “Fml1^ΔC^ and Fml1-MHF catalyse replication fork restoration in vitro” and “Rad52 inhibits unwinding of a model IFSA junction by Fml1^ΔC^ and Fml1-MHF”, with further discussion on in paragraph five of the Discussion section.

3) Subsection “Identification of a mutation in Mhf2 that disrupts the interaction between Fml1 and MHF” and Figure 3B: the epistasis claimed is not supported by these data, since the mhf2^D87A^ mutant shows no sensitivity to MMS at these doses (and therefore epistasis can't be measured). This experiment should be done at 0.01% MMS, where it is sensitive.

We have repeated the experiment in Figure 3B to include higher doses of MMS. The new data support the epistasis claim.

Major points that need be addressed in revision:

*1)* (Figure 6) *The authors compare biochemical activities of Fml1-MHF and Fml1 deltaC. The lagging gap and regressed forks are the most likely substrates* in vivo *and both are efficiently restored by Fml1-MHF. Fml1C exhibits higher activity than Fml1-MHF on the lagging gap substrate at sub-stoichiometric concentration, but at stoichiometric concentration, it restores the fork more slowly than Fml1-MHF. Given the high activity of Fml1delta C on the lagging strand substrate, it is surprising that it is not very effective in suppressing deletions. Is Fml1delta C recruited to the stalled fork?*

We suspect that Fml1^∆C^’s recruitment to collapsed forks is impaired but we currently don’t have any direct evidence for this. However, we have extended our Discussion to mention that Fml1 might require its C-terminal domain (and interaction with MHF) to be efficiently recruited to collapsed replication forks.

2) Abstract: "enhanced by Mhf1 and Mhf2" this claim is too strong since the fork restoration activity of Fml1 in the absence of Mhf1/2 isn't done.

We have removed this statement.

3) Subsection “Fml1 suppresses Rad51-independent SDDs”: it does look like there are some Rad51-dependent deletions as well, please comment

We can detect a reduction in deletions in a *rad51*∆ mutant when there is no DNA spacer between the *ade6*^-^ repeats (e.g. see Sun et al., 2008) and when the spacer is only 2 kb (Figure 1). However, with a 5 kb spacer we don’t see a reduction (Morrow et al., 2017). We have added the following text to make it clear that some deletions can be formed by Rad51: “In a rad51∆ single mutant, the frequency of SDDs is ~2-fold less than in wild-type (p < 0.0001) indicating that some deletions are formed by Rad51.” We also mention in the Discussion that Fml1 may limit Rad51-dependent deletions by its D-loop unwinding activity.

4) Subsection “Identification of a mutation in Mhf2 that disrupts the interaction between Fml1 and MHF” paragraph two: It is surprising by the extent of mobility change conferred by the D87A Mhf2 mutant. It is interpreted as being a subtle, separation-of-function mutant, but is seems likely there is a gross conformational change. This should be acknowledged.

We have added the following text to the legend for Figure 3—figure supplement 2A: “Note that His-tagged Mhf2^D87A^ co-migrates with Mhf1 unlike His-tagged Mhf2, which exhibits slower migration. Proteins were fully denatured by boiling in SDS loading dye containing ß-mercaptoethanol prior to running on the gel. Therefore, the difference in migration of His-tagged Mhf2^D87A^ and His-tagged Mhf2 is not due to a difference in protein conformation.”

The size-exclusion chromatography profiles of Mhf1-HisMhf2 and Mhf1-HisMhf2^D87A^ are also identical, which suggests that the D87A mutation does not cause a major change in protein conformation. One possible explanation for the difference in mobility of His-Mhf2 and His-Mhf2^D87A^ is that the D to A change increases hydrophobicity, which is known to promote faster migration in SDS-PAGE (see: Shirai et al., 2008, J. Biol. Chem. 283: 10745-10752).

5) Paragraph three in the same section: "speckled" this looks diffuse to me plus the signal from Mhf1 is more extensive than the DAPI. Explain?

We have changed “speckled” to “diffuse”. We suspect that the diffuse Mhf-GFP fluorescence that extends beyond the DAPI staining is in the nucleolus (which is not stained by DAPI). This is now mentioned in the Figure 3—figure supplement 3B legend.

6) Discussion in general: Is there some consequence/advantage of recruiting Fml1 to the centromere? Or do the authors think that Fml1 only plays a role at Mhf1/2 binding to noncentromeric sites?

We have no evidence that Fml1 is preferentially recruited to centromeres compared to non-centromeric sites, and the data in Figure 3—figure supplement 3B show that, unlike its non-centromeric localisation, Mhf1-GFP’s ability to localise to centromeres is independent of Fml1. Moreover, in unpublished work we have found that MHF, when bound to its centromeric partner proteins (CENP-T and CENP-W), is unable to interact with Fml1. Nevertheless, it has been reported by others that Fml1 does function at centromeres to suppress gross chromosomal rearrangements caused by recombination between centromeric repeats (Zafar et al., 2017, Nucleic Acids Research 45: 11222-11235). However, at the moment we don’t have anything further to add to the discussion on whether Fml1’s activity at centromeres is fundamentally different from its activity at non-centromeric sites.